# A Sound Definitional Interpreter for a Simply Typed Functional Language

**Burak Ekici** 

Department of Computer Engineering, Muğla Sıtkı Koçman University, Muğla 48000, Turkey; burakekici@mu.edu.tr

**Abstract:** In this paper, we develop, in the proof assistant *Coq*, a definitional interpreter and a type-checker for a simply typed functional language, and formally prove that the mentioned type-checker is sound with respect to the definitional interpreter via progress and preservation. To represent binders, we embark on the choice of "concrete syntax" in which parameters are just names (or strings).

**Keywords:** definitional interpreters; simply typed functional languages; formal soundness proofs; the Coq proof assistant



## 1. Introduction

In the domain of programming language semantics, interpreters [1] constitute the bases for the abstract specification of higher-order programming languages. This specification is simply handled by defining the semantics of an object language in terms of the semantics of a target (or host) language. An interpreter is said to be definitional if it is computational; that is, if it could be executed to test and validate the semantics of the object language programs in the "trusted" target language with well-known semantics. Functional programming languages that support algebraic data types (such as Haskell, OCaml, Coq, etc.), especially those serving dependent types, are suitable targets to implement interpreters. We definitionally formalize small-step operational semantics of a simply typed functional language, a variant of simply typed Lambda calculus ($\lambda^{\rightarrow}$) [2], alongside a type-checker in the Coq proof assistant [3]. In such an approach, one needs to bridge the interpreter and the type-checker together simply by proving that the type system is sound, potentially embarking on the approach proposed by Wright and Felleisen [4]. In more formal terms, one needs to make sure that the following properties hold.

(i)   If an arbitrarily given term is well-typed, then it either reduces in a single step into some other term or it is already a value. That is, well-typed terms are never stuck;

(ii)  the type of a given term does not vary under reduction/evaluation.

The former property is called progress while the latter is known as preservation. Putting all related technical machinery together and obtaining these proofs even for a not-so-rich object language might very well be tedious and, thus, error-prone. This underpins the main target of the paper, where we carefully investigate and prove the properties (i) and (ii) for a simply typed functional language (extending $\lambda^{\rightarrow}$) both in a pen-and-paper setting and in a Coq formalization. We detail and present technical machinery that constructs those proofs in both contexts and relate one to the other. To the best of the author's knowledge, such a Coq development does not exist out there with a similar extent of technicality (as well as the approach). Related formalization is not definitional and does not prove the soundness via (i) and (ii).

*1.1. Related Work and Contributions*

The approach employed by [5–8] benefits from intrinsically typed abstract syntax definitions when it comes to encoding the object language terms into a dependently typed target language. This way, the type checker of the target language is enabled to automatically verify the safety type of the interpreter. The approach has been extended in [9] such that it supports linearly typed languages as well.

Darais et al. [10,11] studied abstract definitional interpreters written in a monadic style, which makes it possible to capture operations that interact with the outside world. Therefore, languages with impure features (e.g., use of local/global state, mechanisms to handle exceptions, etc.) could also be nicely interpreted. This approach has close relations with that of [12–16] where definitional interpreters mimic state machines, especially nondeterministic pushdown automata.

With all of these mentioned, the main contribution of this paper lies in the Coq formalization of an interpreter for a simply typed language alongside its soundness proof captured by progress and preservation properties. In these lines, novelties are a 'few folded', as mentioned below:

- The presented interpreter is implemented to be computational; not living in Coq's `Prop`. Amin et al. [17] definitionally implemented a similar interpreter; however, the attached soundness proof was not handled by progress and preservation. Moreover, the approach conducted in Software Foundations [18] and Koprowski's paper [19] makes use of non-definitional interpreters, living in Coq's `Prop`, it, however, obtains soundness proof via progress and preservation. In essence, we combine these approaches.
- We aimed to have complete literature on the Coq formalization surrounding the definitional interpreters for simply typed languages, alongside [20,21]. We discuss the technical machinery that closes the soundness proof of the interpreter. Even though the main message of the paper is well-known, to the author's best knowledge, there is no Coq formalization that implements both type-checking and reduction (for some extension of $\lambda^{\rightarrow}$) in a definitional fashion (outside Coq's `Prop`), formally relating to one another.
- Our $\lambda^{\rightarrow}$ extension involves a 'fixpoint', branching constructors, and a few binary operations over natural numbers and Booleans.

The Coq formalization already deals with an extension of $\lambda^{\rightarrow}$ interpreted as a case study. We could make use of the same approach presented here (https://github.com/ekiciburak/Lambda2, accessed on 8 December 2022) (a definitional interpreted for a polymorphically typed functional language coded in Haskell), enriching the current state of our interpreter. The extension is very easily adaptable, as discussed in [17]; it brings over new cases as part of the soundness and will be targeted in the near future.

In this regard, the milestones of the formalization are explained in Sections 2.3, 2.4 and 3 while the whole library is accessible at https://github.com/ekiciburak/extSTLC/tree/master (accessed on 17 December 2020). For the file organization of the library, please refer to Appendix A.

*1.2. Organization of the Paper*

In Section 2, we first give a quick recap of the untyped Lambda calculus. We then present an extension with natural numbers, Booleans, branching, and fixpoint constructors. Afterward, we introduce a definitional interpreter based on small-step call-by-value semantics, and a type-system employing simple (or non-dependent) types. Along the same lines, we give a taste of such functions in a Coq formalization. In Section 3, we discuss, in deep technical details, the soundness of the type system with respect to the interpretation via both pen-and-paper and Coq-certified proofs.

## 2. A Quick Recap of the $\lambda$ Calculus

The $\lambda$ calculus is a mathematical formalism that expresses computations as function evaluations. It treats functions in an *intensional* viewpoint in which a function simply is

an abstraction composed of input arguments and a body. The equality over functions is syntactically tested in between function bodies modulo renaming (*α-equivalence* or renaming) of the bound arguments. The principle of *functional extensionality* cannot be directly proven there but can be taken as an axiom whenever needed. The functions with multiple input arguments can be rewritten as sequences of functions using a single argument. This technique is known as *Currying* or *Schonfinkeling*. We now state below the syntax of the $\lambda$ calculus in an inductive fashion:

$$
\begin{array}{llll}
t & := & x & \text{term variable/identifier} \\
 & | & \lambda x.\,t & \text{function abstraction} \\
 & | & t\,t & \text{function application}
\end{array}
$$

An abstraction $\lambda x.\,t$ is indeed a representation of a function with the *bound* input argument $x$ and the body $t$ separated by the period '.' symbol. For instance, thinking of a function $f(a) := a + 1$, with argument $a$ ranging over natural numbers, one can simply encode it as $\lambda a.\,a\ +1$ within the Lambda terms. Here, the Greek letter $\lambda$ is chosen to anonymously name functions as it does not matter whether to name functions. Thus, all of them are uniquely named $\lambda$.

To express a computation within the scope of $\lambda$ calculus, the function applications of the form $t\,t$ are employed. It is not possible to evaluate terms under the binder Lambda. Namely, an abstraction alone cannot be evaluated. It is necessary to have an abstraction applied to a term (*reducible term* or *redex*) when it comes to speaking of function evaluations:

$$(\lambda x.\,M)\,N \rightarrow_\beta M[x := N]$$

This one step of the evaluation is known as *β-reduction* and is denoted by the right-arrow with the letter $\beta$ sub-scripted: '$\rightarrow_\beta$'. The $\beta$-reduction can be viewed as a single step of the computation in which an application of the form $(\lambda x.\,M)\,N$ evaluates or reduces into a term $M[x := N]$ denoting a variant of $M$ in which every occurrence of the argument $x$ is substituted by the applicant term $N$.

**Example 1.** *The term $(\lambda x.\,x + y)\,10$ evaluates in one beta-step into $10 + y$. Notice that the variable $y$ here is not bound by the binder Lambda. These kinds of terms in Lambda expressions are known as free variables.*

**Example 2** (Church Encoding). *It is possible to encode natural numbers in $\lambda$ calculus in such a way that the term $\lambda f.\,\lambda n.\,f\,n$ denotes the natural number 1 while $\lambda f.\,\lambda n.\,f\,f\,f\,n$ denotes the natural 3. Namely, the number of applications of the term $f$ over the term $n$ defines the corresponding natural number.*

**Example 3.** *Let $t := \lambda x.\,\lambda y.\,x$ in $1\,t = (\lambda f.\,\lambda n.\,f\,n)\,t \rightarrow_\beta \lambda n.\,t\,n = \lambda n.\,(\lambda x.\,\lambda y.\,x)\,n \rightarrow_\beta \lambda n.\,(\lambda x.\,\lambda y.\,x)\,n$.*

**Example 4.** *Let $\Omega := \lambda x.\,x\,x$ in $\Omega\,\Omega = \lambda x.\,x\,x\,(\lambda x.\,x\,x) \rightarrow_\beta \Omega\,\Omega \rightarrow_\beta \Omega\,\Omega \rightarrow_\beta \Omega\,\Omega \dots$*

**Example 5.** *There are two ways to beta-reduce the term $\Omega\,1\,t$: (i) into $1\,1\,t$ if the application on the left is accounted for first; (ii) into $\Omega\,\lambda n.\,t\,n$ if the one on the right.*

Notice that the expression $\Omega\,\Omega$ in Example 4 loops under beta-reduction returning itself. Such an evaluation never reaches a final or termination state. Moreover, the Example 5 aims at presenting the fact that non-deterministic evaluations can also be encoded within the scope of $\lambda$ calculus. The goal to have them is to informally show that $\lambda$ calculus is indeed Turing complete. Namely, any computation that could be simulated by a Turing machine could also be implemented within Lambda terms. This conclusion can also be inferred from the famous Church–Turing Thesis but cannot be formally proven as it is not mathematically stated. However, one can disprove it just by refutation. Namely, it suffices

for one to come up with a computation that can be simulated with one model but not with the other.

### 2.1. Evaluation Strategies

In order to impose an order of evaluation for a deterministic reduction of Lambda terms, several strategies are put forward. We discuss in this section, the strategy named *call-by-value* (CBV) and skip the others as it constitutes a basis for our Coq development. The CBV permits an application to reduce only after reducing its argument into a *value*. Namely, considering the term $e_1\,e_2$, CBV ensures that $e_2$ is fully evaluated until no further reduction steps are possible (into the normal form), and then the application takes place. That could formally be stated as:

$$\frac{\texttt{isvalue }e_2 = \texttt{true}}{(\lambda x.\,e_1)\,e_2 \to_\beta e_1[x := e_2]}\;(app_1) \qquad \frac{\texttt{isvalue }e_2 = \texttt{false} \quad e_2 \to_\beta e_2'}{(\lambda x.\,e_1)\,e_2 \to_\beta (\lambda x.\,e_1)\,e_2'}\;(app_2)$$

such that the `isvalue` and `substitution` (denoted $?[? := ?]$) functions are formally given in Definitions 1 and 2, respectively.

**Definition 1.** *The* `isvalue` *function is defined as follows:*

$$\begin{aligned}
\texttt{isvalue }(\lambda x.\,e) &\Rightarrow & true \\
\texttt{isvalue }\_ &\Rightarrow & false
\end{aligned}$$

**Definition 2.** *The substitution function is recursively implemented as follows:*

$$\begin{aligned}
Ident\;x[s := v] &\Rightarrow & \text{if } x = s \text{ then } v \text{ else } Ident\;x \\
(\lambda x.\,e)[s := v] &\Rightarrow & \text{if } x \neq s \text{ then } \lambda x.\,e[s := v] \text{ else } \lambda x.\,e
\end{aligned}$$

### 2.2. Extensions

For a more involved calculus, let us now extend the set of Lambda terms, and accordingly, the accompanying evaluation strategy. We drop in natural numbers, Booleans, a technique to handle branching and a fixpoint combinator. Moreover, we include three operations over natural numbers, addition, subtraction, and multiplication, together with a pair of Boolean comparison operators: equality and greater-than checks. The extended set of terms are lsited in Figure 1:

| $t$ | := | $Ident\;x$ | term variable/identifier |
|-----|----|-----|-----|
| | \| | $\lambda x.\,t$ | function abstraction |
| | \| | $t\,t$ | function application |
| | \| | $NVal\;n$ | natural numbers |
| | \| | $BVal\;b$ | Booleans |
| | \| | $ITE\;t\,t\,t$ | if-then-else branching |
| | \| | $Fix\;t$ | fixpoint combinator |
| | \| | $Plus\;t\,t$ | addition over natural |
| | \| | $Minus\;t\,t$ | subtraction over natural |
| | \| | $Mult\;t\,t$ | multiplication over natural |
| | \| | $Eq\;t\,t$ | equality check over Booleans |
| | \| | $Gt\;t\,t$ | greater-than check over Booleans |

**Figure 1.** Extended Lambda terms.

In Figure 2, we extend the CBV strategy, such that it covers the newly introduced terms as well.

**Definition 3.** *The* `isvalue` *function is updated into:*

$$\begin{aligned}
\texttt{isvalue }(\lambda x.\,e) &\Rightarrow & true \\
\texttt{isvalue }(NVal\;n) &\Rightarrow & true \\
\texttt{isvalue }(BVal\;b) &\Rightarrow & true \\
\texttt{isvalue }\_ &\Rightarrow & false
\end{aligned}$$

$$\frac{\texttt{isvalue } e_1 = \texttt{true} \quad e_2 \rightarrow_\beta e_2'}{e_1\, e_2 \rightarrow_\beta e_1\, e_2'}\ (app_1)$$

$$\frac{\texttt{isvalue } e_1 = \texttt{false} \quad e_1 \rightarrow_\beta e_1'}{e_1\, e_2 \rightarrow_\beta e_1'\, e_2}\ (app_2)$$

$$\frac{\texttt{isvalue } e_2 = \texttt{true}}{(\lambda x.\, e_1)\, e_2 \rightarrow_\beta e_1[x := e_2]}\ (app_3)$$

$$\frac{\texttt{isvalue } e_2 = \texttt{false} \quad e_2 \rightarrow_\beta e_2'}{(\lambda x.\, e_1)\, e_2 \rightarrow_\beta (\lambda x.\, e_1)\, e_2'}\ (app_4)$$

$$\frac{}{\text{Fix } (\lambda x.\, e_1) \rightarrow_\beta e_1[x := \text{Fix } (\lambda x.\, e_1)]}\ (fix_1)$$

$$\frac{f \rightarrow_\beta f'}{\text{Fix } f \rightarrow_\beta \text{Fix } f'}\ (fix_2)$$

$$\frac{}{\text{ITE (BVal true) } e_2\, e_3 \rightarrow_\beta e_2}\ (ite_1)$$

$$\frac{}{\text{ITE (BVal false) } e_2\, e_3 \rightarrow_\beta e_3}\ (ite_2)$$

$$\frac{(e_1 \rightarrow_\beta e_1')}{\text{ITE } e_1\, e_2\, e_3 \rightarrow_\beta \text{ITE } e_1'\, e_2\, e_3}\ (ite_3)$$

$$\frac{}{\text{Plus (NVal } n)\text{ (NVal } m) \rightarrow_\beta \text{NVal } (n + m)}\ (plus_1)$$

$$\frac{\texttt{isvalue } a = \texttt{true} \quad b \rightarrow_\beta b'}{\text{Plus } a\, b \rightarrow_\beta \text{Plus } a\, b'}\ (plus_2)$$

$$\frac{\texttt{isvalue } a = \texttt{false} \quad a \rightarrow_\beta a'}{\text{Plus } a\, b \rightarrow_\beta \text{Plus } a'\, b}\ (plus_3)$$

$$\frac{}{\text{Minus (NVal } n)\text{ (NVal } m) \rightarrow_\beta \text{NVal } (n - m)}\ (minus_1)$$

$$\frac{\texttt{isvalue } a = \texttt{true} \quad b \rightarrow_\beta b'}{\text{Minus } a\, b \rightarrow_\beta \text{Minus } a\, b'}\ (minus_2)$$

$$\frac{\texttt{isvalue } a = \texttt{false} \quad a \rightarrow_\beta a'}{\text{Minus } a\, b \rightarrow_\beta \text{Minus } a'\, b}\ (minus_3)$$

$$\frac{}{\text{Mult (NVal } n)\text{ (NVal } m) \rightarrow_\beta \text{NVal } (n \times m)}\ (mult_1)$$

$$\frac{\texttt{isvalue } a = \texttt{true} \quad b \rightarrow_\beta b'}{\text{Mult } a\, b \rightarrow_\beta \text{Mult } a\, b'}\ (mult_2)$$

$$\frac{\texttt{isvalue } a = \texttt{false} \quad a \rightarrow_\beta a'}{\text{Mult } a\, b \rightarrow_\beta \text{Mult } a'\, b}\ (mult_3)$$

$$\frac{}{\text{Eq (NVal } a)\text{ (NVal } b) \rightarrow_\beta \text{BVal } (a = b)}\ (eq_1)$$

$$\frac{\texttt{isvalue } a = \texttt{true} \quad b \rightarrow_\beta b'}{\text{Eq } a\, b \rightarrow_\beta \text{Eq } a\, b'}\ (eq_2)$$

$$\frac{\texttt{isvalue } a = \texttt{false} \quad a \rightarrow_\beta a'}{\text{Eq } a\, b \rightarrow_\beta \text{Eq } a'\, b}\ (eq_3)$$

$$\frac{}{\text{Gt (NVal } a)\text{ (NVal } b) \rightarrow_\beta \text{BVal } (a > b)}\ (gt_1)$$

$$\frac{\texttt{isvalue } a = \texttt{true} \quad b \rightarrow_\beta b'}{\text{Gt } a\, b \rightarrow_\beta \text{Gt } a\, b'}\ (gt_2)$$

$$\frac{\texttt{isvalue } a = \texttt{false} \quad a \rightarrow_\beta a'}{\text{Gt } a\, b \rightarrow_\beta \text{Gt } a'\, b}\ (gt_3)$$

**Figure 2.** Small-step CBV semantics.

**Definition 4.** *Moreover, the substitution function is extended in the following cases:*

$$
\begin{array}{lcl}
(t_1\, t_2)[s := v] & \Rightarrow & (t_1[s := v])\, (t_2[s := v]) \\
(ITE\, t_1\, t_2\, t_3)[s := v] & \Rightarrow & ITE\, (t_1[s := v])\, (t_2[s := v])\, (t_3[s := v]) \\
(Fix\, t_1)[s := v] & \Rightarrow & Fix\, (t_1[s := v]) \\
(Plus\, t_1\, t_2)[s := v] & \Rightarrow & Plus\, (t_1[s := v])\, (t_2[s := v]) \\
(Minus\, t_1\, t_2)[s := v] & \Rightarrow & Minus\, (t_1[s := v])\, (t_2[s := v]) \\
(Mult\, t_1\, t_2)[s := v] & \Rightarrow & Mult\, (t_1[s := v])\, (t_2[s := v]) \\
(Eq\, t_1\, t_2)[s := v] & \Rightarrow & Eq\, (t_1[s := v])\, (t_2[s := v]) \\
(Gt\, t_1\, t_2)[s := v] & \Rightarrow & Gt\, (t_1[s := v])\, (t_2[s := v]) \\
(NVal\, n)[s := v] & \Rightarrow & NVal\, n \\
(BVal\, b)[s := v] & \Rightarrow & BVal\, b \\
\end{array}
$$

Notice that the symbols '+', '−' and '×' appearing in the conclusions of the rules $plus_1$, $minus_1$, and $mult_1$ are, respectively, denoting addition, subtraction, and multiplication over natural numbers. Similarly, the symbols '=' and '>' used in the conclusions of the rules $eq_1$ and $gt_1$ are employed to denote Boolean equality and greater-than checks. The rules to define branching $ite_1$, $ite_2$, and $ite_3$ are indeed folklore. The term ITE $e_1\ e_2\ e_3$ evaluates in a single step into $e_2$ if $e_1$ is the Boolean value true; into $e_3$ if $e_2$ is the Boolean false. Otherwise, it reduces to ITE $e_1'\ e_2\ e_3$ if $e_1$ evaluates into $e_1'$. There is a fixpoint combinator Fix that takes a term $f$ and, in a single beta-step, replaces every occurrence of the variable (or identifier) $x$ with the term $f$ itself in $e$ if $f$ is a Lambda term, such as $\lambda x.\, e$. Otherwise, if $f$ is evaluated into $f'$ in a single beta-step then the term Fix $f$ reduces into Fix $f'$. The rules governing operations over 'naturals' and Booleans are also very usual. The term Plus $x\ y$ reduces into the natural number $a + b$ (denoted NVal $(a + b)$) if $x$ is a natural number $a$ (NVal $a$) and $y$ is a natural number $b$ (NVal $b$). If this is not the case, plus $x\ y$ evaluates into Plus $x\ y'$ if $x$ is a value; into Plus $x'\ y$, otherwise, given that $x$ reduces to $x'$ and $y$ reduces to $y'$ in a single step. The other rules concerning subtraction, multiplication, equality, and greater-than checks should be read in the same manner as addition. Note also that the terms that are marked values (by the isvalue function) do not evaluate any further, similar to variables (or identifiers).

### 2.3. A Type System with a Coq Implementation

Looking back at Example 3 in Section 2, one could easily notice that applying the constant term 1 to the first projection function *t* makes no sense in a reasonable sort of mathematics. It is possible to make a similar comment for the term $\Omega$ in Example 4: self-application might be decently awkward. Therefore, to avoid these kinds of unintended cases, and to prune them out, one may follow a syntactic approach that categorizes terms according to the *types* of values they compute. This eventually allows for the proof of the fact that there are no more questions about the aforementioned unintended program behaviors. The set of Lambda terms presented in Figure 1, this time, with the typing information injected-in, is listed in Figure 3.

| $\tau$ | := | *Int* | | *t* | := | ... | ... |
|---|---|---|---|---|---|---|---|
| | \| | *Bool* | | | \| | $\lambda x : \tau . t$ | typed function abstraction |
| | \| | $\tau \rightarrow \tau$ | arrow types | | \| | ... | ... |

**Figure 3.** Type-extended Lambda terms.

The Lambda binder is indeed the only constructor where typing information appears within the term declarations. This simply is to well-type the input variable of a given Lambda term in the construction phase. The rest of the terms are indeed the same as they are given before in Figure 1. It is not cumbersome to reflect these into a Coq implementation:

```
Inductive type: Type ≜
  | Int
  | Bool
  | Arrow: type → type → type.

Inductive term: Type ≜
  | Ident : string → term
  | Lambda: string → type → term → term
  | App   : term  → term → term
  | NVal  : nat   → term
  | BVal  : bool  → term
  | ITE   : term  → term → term → term
  | Fix   : term  → term
  | Plus  : term  → term → term
  | Minus : term  → term → term
  | Mult  : term  → term → term
  | Eq    : term  → term → term
  | Gt    : term  → term → term.
```

One crucial point to underline is that the type system here allows only for ordinary function/arrow types leaving aside dependent and polymorphic function types. Namely, it is a variant and an extension of the simply typed Lambda calculus; not as expressive as, other calculi in Barendregt's cube [22], such as System *F* [23], System *Fω* nor *λC* [24]. The type system involves two base types `Int` and `Bool` with the possibility of inductively constructing "`Arrow`" types out of them. Notice also that in implementing binders, we embark on the choice of "concrete syntax" due to Church [2], in which the identifiers of Lambda terms are just strings. This choice definitely requires some extra care when it comes to avoid *variable capture*. See Section 2.4, especially the text following the `subst` function, for a further explanation. Another point to draw attention to is that the term constructor `NVal` adds to the set of terms the natural numbers (`nat`) of Coq. Similarly, the constructor `BVal` enriches the set of terms with the Coq Booleans (`bool`).

Let us now take a closer look into the typing judgments, formally stated in Figure 4, which are supposed to govern term evaluations just by allowing well-typed terms to reduce only.

We assume that there exists some context $\Gamma$ that is interpreted as a list of typing relations, such as x: $\emptyset$, respecting the *shadowing* property. Namely, if the same variable appears more than once in the context, the leftmost appearance is considered: the one on the left *shadows* others. For instance, suppose we have a context $\Gamma$ in the form of $[(x: \emptyset_1) :: (y: \emptyset_2) :: (x: \emptyset_3)]$, we consider the type of the instance x to be $\emptyset_1$ not $\emptyset_3$. Therefore, under some context $\Gamma$ (denoted "$\Gamma \vdash ?$"), to obtain the type of a given variable *x*,

a *lookup* (starting from the left end of the context), denoted $\Gamma(x)$, is employed ($id_t$). Similarly, under some context $\Gamma$ extended with the fact that some variable $x$ ranging over the type $\tau_1$, if it is possible to deduce that some term $e$ is of type $\tau_2$ then the Lambda term $\lambda x \colon \tau. e$ has the arrow type $\tau_1 \to \tau_2$ under the same context $\Gamma$ ($lam_t$), accordingly denoted $\Gamma, x \colon t_1 \vdash e \colon t_2$. To apply an arbitrary term $e_1$ to a term $e_2$, one needs to make sure that the former term is of an arrow type such that the domain of this type matches with the type of the latter. Moreover, the application $e_1\, e_2$ is now an instance of $e_1$'s co-domain type ($app_t$).

If $f$ is a term of some arrow type $\tau \to \tau$, then the term Fix $f$ is thought to be a fix-point $f$, and must be of type $\tau$ ($fix_t$). The term Plus $e_1\, e_2$ is of the base type *Int* only if both $e_1$ and $e_2$ are of type *Int* ($plus_t$). Similarly, the term Eq $e_1\, e_2$ is of the base type *Bool* only if both $e_1$ and $e_2$ are of type *Int* ($eq_t$). Remark also that $\vdash t \colon \tau$ denotes the fact that the term $t$ is of type $\tau$ under the *empty* context. In a Coq formalization, we implement the rules stated in Figure 4, using a recursive function named `typecheck`, as follows:

```
Fixpoint typecheck (m: ctx) (e: term): option type ≜
  match e with
    | Ident s      ⇒ lookup m s
    | NVal n       ⇒ Some Int
    | BVal b       ⇒ Some Bool
    | Lambda x t e1 ⇒ let n ≜ extend m x t in
                      let te1 ≜ typecheck n e1 in
                      match te1 with
                        | Some te1 ⇒ Some (Arrow t te1)
                        | None ⇒ None
                      end
    | App e1 e2    ⇒ let te1 ≜ typecheck m e1 in
                     let te2 ≜ typecheck m e2 in
                     match te1, te2 with
                       | Some (Arrow te1s te1t), Some te2 ⇒
                           if type_eqb te1s te2 then Some te1t else None
                       | _, _  ⇒ None
                     end
    | ITE e1 e2 e3 ⇒ let te1 ≜ typecheck m e1 in
                     let te2 ≜ typecheck m e2 in
                     let te3 ≜ typecheck m e3 in
                     match te1, te2, te3 with
                       | Some nte1, Some nte2, Some nte3 ⇒
                           if type_eqb nte1 Bool && type_eqb nte2 nte3
                           then Some nte2 else None
                       | _, _, _   ⇒ None
                     end
    | Fix f        ⇒ let tf ≜ typecheck m f in
                     match tf with
                       | Some (Arrow t1 t2) ⇒ if type_eqb t1 t2
                                              then Some t2 else None
                       | _                  ⇒ None
                     end
    | Plus a b     ⇒ let t1 ≜ typecheck m a in
                     let t2 ≜ typecheck m b in
                     match t1, t2 with
                       | Some Int, Some Int ⇒ Some Int
                       | _, _               ⇒ None
                     end
    | ...
  end.
```

where the context `ctx` is a list of `string-type` pairs which always is `extended` on the left-hand side to obtain the shadowing property (explained above) satisfied.

```
Definition ctx ≜ list (string * type)%type.
Definition extend (c: ctx) (x: string) (t: type)≜ (x, t):: c.
Fixpoint lookup {A: Type} (c: list (string * A)) (s: string): option A ≜
  match c with
    | nil ⇒ None
    | (x, t):: r ⇒ if eqb x s then Some t else (lookup r s)
  end.
```

In our implementation, `lookups` are always performed by/starting from the left end of a given context. This matches with the idea that designates the shadowing feature. The output type of the `lookup` function is wrapped by the `option` type (or monad [25]) of Coq, just in case that the input variable is absent from the provided context: the function ends up returning `None`.

The function `typecheck` takes a term along with a context, and outputs the type of the term under the input context. Similar to that of the `lookup`, the instances returned by the `typecheck` function are also wrapped with the `option` type so that upon ill-typed inputs, such as `App (NVal 5) (BVal false)`, the function returns `None`.

$$\frac{}{\Gamma \vdash \text{Ident } x \colon \Gamma(x)} \ (id_t) \qquad\qquad \frac{\Gamma, x \colon \tau_1 \vdash e \colon \tau_2}{\Gamma \vdash \lambda x \colon \tau_1 . e \colon \tau_1 \to \tau_2} \ (lam_t)$$

$$\frac{}{\Gamma \vdash \text{NVal } n \colon Int} \ (nval_t) \qquad\qquad \frac{}{\Gamma \vdash \text{BVal } b \colon Bool} \ (bval_t)$$

$$\frac{\Gamma \vdash e_1 \colon \tau_1 \to \tau_2 \quad \Gamma \vdash e_2 \colon \tau_1}{\Gamma \vdash e_1 \, e_2 \colon \tau_2} \ (app_t) \qquad\qquad \frac{\Gamma \vdash f \colon \tau \to \tau}{\Gamma \vdash \text{Fix } f \colon \tau} \ (fix_t)$$

$$\frac{\Gamma \vdash e_1 \colon Bool \quad \Gamma \vdash e_2 \colon \tau \quad \Gamma \vdash e_3 \colon \tau}{\Gamma \vdash \text{ITE } e_1 \, e_2 \, e_2 \colon \tau} \ (ite_t) \qquad\qquad \frac{\Gamma \vdash e_1 \colon Int \quad \Gamma \vdash e_2 \colon Int}{\Gamma \vdash \text{Plus } e_1 \, e_2 \colon Int} \ (plus_t)$$

$$\frac{\Gamma \vdash e_1 \colon Int \quad \Gamma \vdash e_2 \colon Int}{\Gamma \vdash \text{Minus } e_1 \, e_2 \colon Int} \ (minus_t) \qquad\qquad \frac{\Gamma \vdash e_1 \colon Int \quad \Gamma \vdash e_2 \colon Int}{\Gamma \vdash \text{Mult } e_1 \, e_2 \colon Int} \ (mult_t)$$

$$\frac{\Gamma \vdash e_1 \colon Int \quad \Gamma \vdash e_2 \colon Int}{\Gamma \vdash \text{Eq } e_1 \, e_2 \colon Bool} \ (eq_t) \qquad\qquad \frac{\Gamma \vdash e_1 \colon Int \quad \Gamma \vdash e_2 \colon Int}{\Gamma \vdash \text{Gt } e_1 \, e_2 \colon Bool} \ (gt_t)$$

**Figure 4.** Typing judgments.

### 2.4. A Definitional Interpreter in Coq

We first adapt the rules stated in Figure 2, such that the typing information explicitly appears in reduction steps. To do so, it suffices to add types to the terms where the Lambda binder is involved as it is the only constructor that embodies syntactical typing information. Therefore, only the rules $app_3$ and $app_4$ are re-designated and presented in Figure 5.

$$\frac{\text{isvalue } e_2 = \texttt{true}}{(\lambda x \colon \tau . e_1) \, e_2 \to_\beta e_1[x := e_2]} \ (app_3) \qquad \frac{\text{isvalue } e_2 = \texttt{false} \quad e_2 \to_\beta e_2'}{(\lambda x \colon \tau . e_1) \, e_2 \to_\beta (\lambda x \colon \tau . e_1) \, e_2'} \ (app_4)$$

**Figure 5.** Typing-adapted small-step CBV semantics.

A small-step CBV interpreter based on the *typing-adapted* versions of the rules (as in Figure 5) stated in Figure 2 is *definitionally* implemented in Coq as follows:

```
Fixpoint beta (e: term): option term ≜
  match e with
    | Ident s                    ⇒ None
    | Lambda x t e               ⇒ None
    | NVal n                     ⇒ None
    | BVal b                     ⇒ None
    | (Fix (Lambda x t e1)) as f ⇒ Some (subst e1 x f)
    | Fix f                      ⇒ let bf ≜ beta f in
                                     match bf with
                                       | Some sbf  ⇒ Some (Fix sbf)
                                       | None      ⇒ None
                                     end
    | App (Lambda x t e1) e2     ⇒ if isvalue e2 then Some (subst e1 x e2)
                                   else
                                     let e2' ≜ beta e2 in
                                     match e2' with
                                       | Some e2'' ⇒ Some (App (Lambda x t e1) e2'')
                                       | None      ⇒ None
                                     end
    | App e1 e2                  ⇒ if isvalue e1 then
                                     let e2' ≜ beta e2 in
                                     match e2' with
                                       | Some e2'' ⇒ Some (App e1 e2'')
                                       | None      ⇒ None
                                     end
```

```
                                      else
                                        let e1' ≜ beta e1 in
                                        match e1' with
                                          | Some e1'' ⇒ Some (App e1'' e2)
                                          | None     ⇒ None
                                        end
    | ITE (BVal true) e2 e3      ⇒ Some e2
    | ITE (BVal false) e2 e3     ⇒ Some e3
    | ITE e1 e2 e3               ⇒ let e1' ≜ beta e1 in
                                    match e1' with
                                      | Some e1''   ⇒ Some (ITE e1'' e2 e3)
                                      | None        ⇒ None
                                    end
    | Plus (NVal n) (NVal m)     ⇒ Some (NVal (n + m))
    | Plus a b                   ⇒ if isvalue a then
                                      let b' ≜ beta b in
                                      match b' with
                                        | Some b''  ⇒ Some (Plus a b'')
                                        | None      ⇒ None
                                      end
                                    else
                                      let a' ≜ beta a in
                                      match a' with
                                        | Some a''  ⇒ Some (Plus a'' b)
                                        | None      ⇒ None
                                      end
    | ...
  end.
```

where the substitution function subst (Definition 4) is formalized to be:

```
Fixpoint subst (e: term) (x: string) (n: term): term ≜
  match e with
    | Ident s      ⇒ if String.eqb s x then n else e
    | Lambda y t m ⇒ if (Bool.eqb (String.eqb y x) false)
                        then Lambda y t (subst m x n) else e
    | Fix f        ⇒ Fix (subst f x n)
    | App t r      ⇒ App (subst t x n) (subst r x n)
    | ITE t1 t2 t3 ⇒ ITE (subst t1 x n) (subst t2 x n) (subst t3 x n)
    | Plus a b     ⇒ Plus (subst a x n) (subst b x n)
    | ...
  end.
```

The point that this interpreter being definitional stems from the fact that it takes a term, and *computes* in real time a reduced term out of it. This is in fact the point where our interpreter differs from that of 'Software Foundations' [18] in which interpretation is not computational as it is developed in Coq's Prop. The return type of the function beta is wrapped by the option type for stuck (see Definition 6) terms, such as App (NVal 5) (BVal false), the function returns None, meaning no further reduction steps on this term is possible. Notice that we skipped in the above definition the match-cases for operators, such as Minus, Mult, Eq and Gt because they are very similar to that of Plus. Please refer to the accompanying Coq library for the complete definition. Considering the substitution, we implement a function (named subst above) that is capture avoiding but in a restricted form: only to be used in handling beta-reduction of closed terms; namely terms that do not involve free variables. To generalize the implementation so that it also handles terms with free variables requires extra care at the match-case "Lambda y t m":

1.  Substitution would be applicable not only when $y \neq x$ but also when $y \notin \text{fv}(n)$ to avoid the bound variable y being captured by any free variable present in the substituted term n.
2.  If the bound variable y is somehow captured by either of the cases $y = x$ and $y \in \text{fv}(n)$, one possibility to avoid the capture is to introduce a fresh variable and replace it with the bound variable y. Another choice may be to completely discard the naming bound variables by employing de Bruijn indices [26] in the Lambda binder, or alternatively pick a locally nameless strategy [27], or maybe embark on (parametric) higher-order abstract syntax [28]. Opting one of these methods, and adjusting the implementation accordingly is set as a future goal.

### 3. Type Soundness

We devote this section to step-by-step explore the soundness proof of the type system presented in Figure 4. It may be beneficial to first look into the chart in Figure 6 that exhibits the dependency among statements proven further in this section, and then read the proof terms in detail.

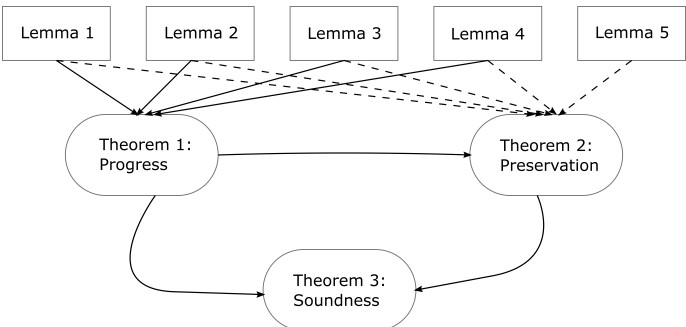

**Figure 6.** Dependency among proof terms.

Remark that the definition, lemma, and theorem names in the upcoming text are indeed links to the corresponding formalization in the Coq library. Please click on the names to browse the related code.

**Lemma 1** (`istypechecked_app`). $\forall t_1\ t_2\ \tau, \ \vdash (t_1\ t_2): \tau \implies \exists v, \vdash t_1: v \to \tau \land \vdash t_2: v$.

**Proof.** The statement is just the inversion of the $(app_t)$ rule presented in Figure 4. Notice that we are trying to show $\exists v, \vdash t_1: v \to \tau \land \vdash t_2: v$ assuming $\vdash (t_1\ t_2): \tau$ (H). The proof proceeds with nested case distinctions over the statements, i.e., whether the terms $t_1$ and $t_2$ are well-typed under the empty context:

1. $\vdash t_1: \kappa \to \tau$ for some arbitrary type $\kappa$.

    (a)  $\vdash t_2: \rho$ for some arbitrary type $\rho$. In this case, both $t_1$ and $t_2$ are individually well-typed under the empty context. By employing this fact, the rule $(app_t)$ and the hypothesis (H), it is easy to deduce that $\kappa = \rho$. Under the same set of assumptions, we could show $\exists v, \vdash t_1: v \to \tau \land \vdash t_2: v$ holds just by plugging in the type $\rho$ (or equivalently $\kappa$) for $v$. The hypothesis (H) would not hold otherwise.

    (b)  $\nvdash t_2: \rho$. Given that the application $(t_1\ t_2)$ is well-typed, under the empty context, the term $t_2$ must be well-typed as well. This case is a violation thus the goal is closed by contradiction.

2. $\nvdash t_1: \kappa \to \rho$. Similar to the item above, if the application $(t_1\ t_2)$ is well-typed, the term $t_1$ must also be well-typed under the empty context context. This case is a violation, thus, the goal is closed by contradiction as well.

    □

**Lemma 2** (`istypechecked_ite`). $\forall t_1\ t_2\ t_3\ \tau, \ \vdash \text{ITE}\ t_1\ t_2\ t_3: \tau \implies \vdash t_1 = Bool \land \vdash t_2: \tau \land \vdash t_3: \tau$.

**Proof.** The statement is the inversion of the rule $(ite_t)$ stated in Figure 4. We aim to prove $\vdash t_1 = Bool \land \vdash t_2: \tau \land \vdash t_3: \tau$ provided $\vdash \text{ITE}\ t_1\ t_2\ t_3: \tau$ (H). The proof we present here proceeds with nested case distinctions over the statements, whether the terms $t_1$, $t_2$ and $t_3$ are well-typed under the empty context:

1. $\vdash t_1: \kappa_1$ for some arbitrary type $\kappa_1$.

    (a)  $\vdash t_2: \kappa_2$ for some arbitrary type $\kappa_2$.

i. $\vdash t_3 : \kappa_3$ for some arbitrary type $\kappa_3$. In this case, the terms $t_1$, $t_2$, and $t_3$ are all individually well-typed under the empty context. With this, the hypothesis ($H$), and the rule ($ite_t$), we could simply deduce the facts that $\kappa_1$ must be *Bool*, and both $\kappa_2$ and $\kappa_3$ must be $\tau$. Any other choice of $\kappa$s would contradict the hypothesis ($H$).

ii. $\nvdash t_3 : \kappa_3$. If the term ITE $t_1$ $t_2$ $t_3$ is well-typed, under the empty context, the term $t_3$ must also be well-typed. This case is a violation, thus, the goal is closed by contradiction.

(b) $\nvdash t_2 : \kappa_2$. The goal in this case similarly holds by contradiction.

2. $\nvdash t_1 : \kappa_1$. The goal in this case is closed by contradiction in a similar manner with that of the above item.

□

**Lemma 3** (`istypechecked_fix`). $\forall t\ \tau, \vdash$ Fix $t : \tau \implies \vdash t : \tau \to \tau$

**Proof.** The statement is the inversion of the rule ($fix_t$) stated in Figure 4. The goal here is to show that $\vdash t : \tau \to \tau$ holds, assuming Fix $t : \tau$ ($H$). The proof proceeds with a case distinction over the statement whether the term $t$ is well-typed under the empty context:

1. $\vdash t : \kappa$ for some arbitrary type $\kappa$. Thanks to the hypothesis ($H$) and the rule ($fix_t$), we could easily reason that $\kappa$ must be $\tau \to \tau$; the hypothesis ($H$) would not hold otherwise.

2. $\nvdash t : \kappa$. If the term Fix $t$ is well-typed, under the empty context, the term $t$ must also be well-typed. This case is a violation, thus, the goal is closed by contradiction.

□

**Lemma 4** (`istypechecked_plus`). $\forall t_1\ t_2\ \tau, \vdash$ Plus $t_1\ t_2 : \tau \implies \tau = Int \wedge \vdash t_1 : Int \wedge \vdash t_2 : Int$

**Proof.** The statement is indeed the inversion of the rule ($plus_t$) in Figure 4. We aim to prove $\tau = Int \wedge \vdash t_1 : Int \wedge \vdash t_2 : Int$ provided that $\vdash$ Plus $t_1\ t_2 : \tau$ ($H$). The proof we present here proceeds with nested case distinctions over the statements, i.e., whether the terms $t_1$ and $t_2$ are well-typed under the empty context:

1. $\vdash t_1 : \kappa_1$ for some arbitrary type $\kappa_1$.

(a) $\vdash t_2 : \kappa_2$ for some arbitrary type $\kappa_2$. In this case, the terms $t_1$ and $t_2$ are individually well-typed under the empty context. With this, the hypothesis ($H$), and the rule ($plus_t$), it is easy to deduce the fact that the types $\tau$, $\kappa_1$ and $\kappa_2$ must be *Int*. Any other choice of $\kappa$s and $\tau$ would be in contradiction with the hypothesis ($H$).

(b) $\nvdash t_2 : \kappa_2$. If the term Plus $t_1\ t_2$ is well-typed, under the empty context, the term $t_2$ must also be well-typed. This case is a violation, thus, the goal is closed by contradiction.

2. $\nvdash t_1 : \kappa_1$. The goal in this case is closed by contradiction in a similar manner with that of the above item.

□

The progress statement claims that if an arbitrary term $t$ type-checks under the empty context then it either reduces in a single step into some term $t'$ or it is already a value. That is, well-typed terms are never *stuck* (see Definition 6 for a formal explanation of being stuck for a term). The terms that are well-typed and do not reduce at the same time are those marked *value*s (see Definition 3).

**Theorem 1** (Progress). $\forall t, \vdash t : \tau \implies$ isvalue $t = true \vee \exists t',\ t \to_\beta t'$ *for some type* $\tau$.

**Proof.** The proof proceeds with structural induction on the term $t$. The cases with Ident $x$, $\lambda x \colon \tau . t$, NVal $n$ and BVal $b$ are indeed trivial: the first assumes false by $\vdash$ Ident $x \colon \tau$ for some type $\tau$, and the rest are already values.

1.  The case with the application $(t_1 \ t_2)$, for some terms $t_1$ and $t_2$, is more involved. There, we aim to prove that $\texttt{isvalue} \ (t_1 \ t_2) = true \vee \exists t', (t_1 \ t_2) \to_\beta t'$ given $\vdash$ $(t_1 \ t_2) \colon \tau$ (*Htc*) for some type $\tau$, along with two induction hypotheses $\vdash t_1 \colon \tau_1 \implies$ $\texttt{isvalue} \ t_1 = true \vee \exists t'_1, t_1 \to_\beta t'_1$ (*IHt$_1$*) for some type $\tau_1$, and $\vdash t_2 \colon \tau_2 \implies$ $\texttt{isvalue} \ t_2 = true \vee \exists t'_2, t_2 \to_\beta t'_2$ (*IHt$_2$*) for some type $\tau_2$. Notice that $\texttt{isvalue}$ $(t_1 \ t_2) = false$; therefore, we consider showing the right disjunction $\exists t', (t_1 \ t_2) \to_\beta t'$ here. It is easy to demonstrate that the Boolean function $\texttt{isvalue}$ is decidable. That is, $\forall t, \texttt{isvalue} \ t = true \ \vee \ \texttt{isvalue} \ t = false$. We start off with a case analysis, specializing this fact on the term $t_1$, and throw two individual subgoals to close the assuming $\texttt{isvalue} \ t_1 = true$ and $\texttt{isvalue} \ t_1 = false$, independently:

    (a)  $\texttt{isvalue} \ t_1 = true$. We proceed with a case analysis specializing the decidability of the $\texttt{isvalue}$ function, this time with the term $t_2$ and obtain two more subgoals to prove, given $\texttt{isvalue} \ t_2 = true$ and $\texttt{isvalue} \ t_2 = false$ separately:

        i.  $\texttt{isvalue} \ t_2 = true$. On a case analysis over the term $t_1$, we in fact have to prove the below three cases in which $t_1$ is a value (other cases, such as $t_1$ being ITE $e_1 \ e_2 \ e_3$, are trivial just because $t_1$s are not values that yield in contradictory cases):

            •   $t_1 = \lambda x \colon \tau_x . e$, for some type $\tau_x$. Mind that the goal here turns out to be $\exists t', (\lambda x \colon \tau_x \ e) \ t_2 \to_\beta t'$. There obviously exists some $t'$; that is the substitution $e[x := t_2]$, closing the goal thanks to the rule ($app_3$) in Figure 2.
            •   $t_1 = $ NVal $n$. In this case, the hypothesis (*Htc*) takes the following shape: $\vdash$ (NVal $n$) $t_2 \colon \tau$, which is indeed false and yields in a contradiction as the first term in an application needs to be of some *arrow*-type but it is of type *Int* here.
            •   $t_1 = $ BVal $b$. This case is proven in a similar fashion to that of the above item.

        ii.  $\texttt{isvalue} \ t_2 = false$. Thanks to Lemma 1, we have $\vdash t_2 \colon \tau_2$, for some type $\tau_2$ out of $\vdash (t_1 \ t_2) \colon \tau$. We use this fact to specialize the induction hypothesis (*IHt$_2$*) to turn it into $\texttt{isvalue} \ t_2 = true \vee \exists t', t_2 \to_\beta t'$. We destruct this, and are supposed to prove the below statements, assuming $\texttt{isvalue} \ t_2 = true$ and $\exists t'_2, t_2 \to_\beta t'_2$ individually:

            A.  $\texttt{isvalue} \ t_2 = true$. This case holds by contradiction.
            B.  $\exists t'_2, t_2 \to_\beta t'_2$. On a case analysis over the term $t_1$, we in fact have to prove the below three cases in which $t_1$ is a value (other cases, such as $t_1$ being ITE $e_1 \ e_2 \ e_3$, are trivial just because $t_1$s are not values that yield in contradictory cases):

                •   $t_1 = \lambda x \colon \tau_x . e$, for some type $\tau_x$. The goal we aim to show in this case is $\exists t', (\lambda x \colon \tau_x . e) \ t_2 \to_\beta t'$. Such a $t'$ obviously exists as $(\lambda x \colon \tau_x . e) \ t'_2$ due to the rule ($app_4$) in Figure 2.
                •   $t_1 = $ NVal $n$. In this case, the hypothesis (*Htc*) takes the following shape: $\vdash$ (NVal $n$) $t_2 \colon \tau$, which is a contradiction, because in an application, the first term needs to be of some *arrow*-type, but it is *Int* here.
                •   $t_1 = $ BVal $b$. This case is proven in a similar manner to that of the above item.

    (b)  $\texttt{isvalue} \ t_1 = false$. Thanks to Lemma 1, we have $\vdash t_1 \colon \tau_1$, for some type $\tau_1$ out of $\vdash (t_1 \ t_2) \colon \tau$. We use this fact to specialize the induction hypothesis (*IHt$_1$*) to turn it into the shape $\texttt{isvalue} \ t_1 = true \vee \exists t', t_1 \to_\beta t'$. We destruct this,

and are supposed to prove the below statements, assuming `isvalue` $t_1 = true$ and $\exists t_1'$, $t_1 \rightarrow_\beta t_1'$ separately:

i.     `isvalue` $t_1 = true$. The statement in this case trivially holds by contradiction.

ii.     $\exists t_1'$, $t_1 \rightarrow_\beta t_1'$. In a case analysis over the term $t_1$, we in fact have to prove subgoals in which $t_1$ is not a value (other cases such as $t_1$ being $\lambda x \colon \tau . e$ are trivial just because $t_1$s are values that yield in proofs by contradiction):

- Recall that we are trying to show $\exists t'$, $(t_1\, t_2) \rightarrow_\beta t'$ such that $t_1$ is not a value. We close all such cases uniformly: there definitely exists a $t'$ being $(t_1'\, t_2)$ given $t_1 \rightarrow_\beta t_1'$, due to the rule ($app_2$) in Figure 2.

2. Looking into the case with ITE $t_1\, t_2\, t_3$, we need to show `isvalue` ITE $t_1\, t_2\, t_3 = true$ $\vee\ \exists t'$, ITE $t_1\, t_2\, t_3 \rightarrow_\beta t'$ given $\vdash$ ITE $t_1\, t_2\, t_3 \colon \tau$ (*Htc*) for some type $\tau$, along with three induction hypotheses $\vdash t_1 \colon \tau_1 \implies$ `isvalue` $t_1 = true \vee \exists t_1'$, $t_1 \rightarrow_\beta t_1'$ (*IHt$_1$*) for some type $\tau_1$, $\vdash t_2 \colon \tau_2 \implies$ `isvalue` $t_2 = true \vee \exists t_2'$, $t_2 \rightarrow_\beta t_2'$ (*IHt$_2$*) for some type $\tau_2$, and $\vdash t_3 \colon \tau_3 \implies$ `isvalue` $t_3 = true \vee \exists t_3'$, $t_3 \rightarrow_\beta t_3'$ (*IHt$_3$*) for some type $\tau_3$. Notice that the sole case we are supposed to demonstrate is $\exists t'$, ITE $t_1\, t_2\, t_3 \rightarrow_\beta t'$ as `isvalue` ITE $t_1\, t_2\, t_3 = false$. Plugging the hypothesis (*Htc*) into Lemma 2, we deduce the facts that $\vdash t_1 \colon$ *Bool* (*Ha*), $\vdash t_2 \colon \tau$ (*Hb*), and $\vdash t_3 \colon \tau$ (*Hc*). We then specialize the induction hypothesis (*IHt$_1$*) with (*Ha*), and obtain `isvalue` $t_1 = true \vee \exists t_1'$, $t_1 \rightarrow_\beta t_1'$. Destructing this hypothesis, we are supposed to prove the goal $\exists t'$, ITE $t_1\, t_2\, t_3 \rightarrow_\beta t'$ twice, assuming `isvalue` $t_1 = true$ and $\vee\ \exists t_1'$, $t_1 \rightarrow_\beta t_1'$ individually:

   (a)     `isvalue` $t_1 = true$. On a case analysis over the term $t_1$, we in fact have to prove the below three cases in which $t_1$ is a value (other cases such as $t_1$ being *Fix f* are trivial just because $t_1$s are not values that yield in contradictory cases):

   - $t_1 = \lambda x \colon \tau_x . e$, for some type $\tau_x$. It is possible to show that $\exists t'$, ITE $(\lambda x \colon \tau_x . e)$ $t_2\, t_3 \rightarrow_\beta t'$ holds just by contradiction as the hypothesis (*Ha*), namely $\vdash (\lambda x \colon \tau_x . e) \colon$ *Bool* proves `False`. The Lambda terms are of the *arrow*-type.
   - $t_1 = $ NVal $n$. In this case, it is similarly possible to prove `False` within the set of hypotheses: considering the hypothesis (*Ha*), namely $\vdash$ NVal $n \colon$ *Bool*, we deduce `False` as it in fact is $\vdash$ NVal $n \colon$ *Int*.
   - $t_1 = $ BVal $b$. If the Boolean variable $b$ is *true*, then the statement $\exists t'$, ITE (BVal *true*) $t_2\, t_3 \rightarrow_\beta t'$ could easily be proven by plugging the term $t_2$ in place of $t'$ due to the rule (*ite$_1$*). If $b$ is the Boolean *false*, then $t'$ is chosen to be $t_3$ to close the goal due to the rule (*ite$_2$*) in Figure 2.

   (b)     $\exists t_1'$, $t_1 \rightarrow_\beta t_1'$. The term $t_1$ is obviously not a value as it reduces at least one step. Therefore, it cannot be BVal $b$, NVal $n$, and $\lambda x \colon \tau_x . e$. We prove the statement $\exists t'$, ITE $t_1\, t_2\, t_3 \rightarrow_\beta t'$ uniformly for the other choices of $t_1$ as follows: just plug in the term ITE $t_1'\, t_2\, t_3$ for the term $t'$ within the goal, and obtain ITE $t_1\, t_2\, t_3 \rightarrow_\beta$ ITE $t_1'\, t_2\, t_3$, which trivially holds thanks to the rule (*ite$_3$*) presented in Figure 2.

3. For the case with Fix $f$, we need to prove that `isvalue` Fix $f = true \vee \exists t'$, Fix $f \rightarrow_\beta t'$ given $\vdash$ Fix $f \colon \tau$ (*Htc*) for some type $\tau$, along with the induction hypothesis $\vdash f \colon \tau \implies$ `isvalue` $f = true \vee \exists f'$, $f \rightarrow_\beta f'$ (*IHf*). Lemma 3 gives proof of the fact that $\vdash f \colon \tau_f$, for some type $\tau_f$ out of (*Htc*). We specialize in the induction hypothesis (*IHf*) with this fact and handle `isvalue` $f = true \vee \exists f'$, $f \rightarrow_\beta f'$. By destructing this, we are supposed to prove the goal $\exists t'$, Fix $f \rightarrow_\beta t'$ (as `isvalue` Fix $f = false$) twice for both `isvalue` $f = true$ and $\exists f'$, $f \rightarrow_\beta f'$.

   (a)     `isvalue` $f = true$. On a case analysis over the term $f$, we in fact have to prove the below three cases in which $f$ is a value (other cases, such as $f$ being

Plus $t_1$ $t_2$ are trivial just because $f$s are not values that yield in contradictory cases):

- $f = \lambda x \colon \tau_x. e$, for some type $\tau_x$. It suffices to plug the term $e[x := \text{Fix} (\lambda x \colon \tau_x. e)]$ into the formula $\exists t'$, Fix $(\lambda x \colon \tau_x. e) \to_\beta t'$, and then apply the rule ($fix_1$) in Figure 2 to have this case proven.

- $f = \text{NVal } n$. In this case, it is possible to prove `False` within the current context: considering ($Ha$), namely $\vdash \text{Fix} (\text{NVal } n) \colon \tau$ is simply false as it is an ill-typed term according to the rules in Figure 4. See the rule ($fix_t$) that states that terms of the form Fix $f$ are well-typed only when $f$ is of some *arrow*-type with the same domain and co-domain. This does not match with the fact that $\vdash \text{NVal } n \colon Int$.

- $f = \text{BVal } b$. This case holds due to the same reason as that of $t_1 = \text{NVal } n$ given above.

(b) $\exists f'$, $f \to_\beta f'$. Given the rule ($fix_2$) in Figure 2, one can reason that $\exists t'$, Fix $f \to_\beta t'$ holds simply by plugging the term Fix $f'$ in, for $t'$ except for the cases where $f$ is a value. These cases are contradictory as there is no $f'$ into which $f$ evaluates.

4. Considering the case involving Plus $t_1$ $t_2$, we need to show isvalue Plus $t_1$ $t_2 = true$ $\vee$ $\exists t'$, Plus $t_1$ $t_2 \to_\beta t'$ given $\vdash$ Plus $t_1$ $t_2 \colon \tau$ ($Htc$) for some type $\tau$, along with two induction hypotheses $\vdash t_1 \colon \tau_1 \implies$ isvalue $t_1 = true \vee \exists t'_1$, $t_1 \to_\beta t'_1$ ($IHt_1$) for some type $\tau_1$, and $\vdash t_2 \colon \tau_2 \implies$ isvalue $t_2 = true \vee \exists t'_2$, $t_2 \to_\beta t'_2$ ($IHt_2$) for some type $\tau_2$. Notice that we are supposed to prove only $\exists t'$, Plus $t_1$ $t_2 \to_\beta t'$ as isvalue Plus $t_1$ $t_2 = false$. Employing Lemma 4, we deduce the facts that $\vdash t_1 \colon Int$ ($Ha$), and that $\vdash t_2 \colon Int$ ($Hb$) out of the hypothesis ($Htc$). Specializing the induction hypotheses ($IHt_1$) with ($Ha$) and ($IHt_2$) with ($Hb$), we obtain isvalue $t_1 = true$ $\vee \exists t'_1$, $t_1 \to_\beta t'_1$ and isvalue $t_2 = true \vee \exists t'_2$, $t_2 \to_\beta t'_2$. We carry on with a case analysis on the decidability of the isvalue function parameterized by the term $t_1$. We, therefore, need to prove the aforementioned goal twice for two distinct cases with isvalue $t_1 = true$ and isvalue $t_1 = false$:

(a) isvalue $t_1 = true$. Applying a case analysis on the term $t_1$, we are supposed to prove the below three cases (others, such as $t_1$ being Fix $f$, are trivial just because $t_1$s are not values that yield in contradictions):

    i. $t_1 = \lambda x \colon \tau_x. e$, for some type $\tau_x$. Notice that in this case, the goal turns out to be $\exists t'$, Plus $(\lambda x \colon \tau_x. e)$ $t_2 \to_\beta t'$. This holds by contradiction just because the type of $\lambda x \colon \tau_x. e$ is an *arrow*-type while it is expected to be *Int* by the hypothesis ($Ha$). Please check out the rule ($lam_t$) given in Figure 4 for a justification.

    ii. $t_1 = \text{NVal } n$. In this case, we start off destructing the induction hypothesis ($IHt_2$), and are in the need of proving the goal $\exists t'$, Plus NVal $n$ $t_2 \to_\beta t'$ twice provided isvalue $t_2 = true$ and $\exists t'_2$, $t_2 \to_\beta t'_2$ individually:

        A. isvalue $t_2 = true$. We proceed with the case analysis of the term $t_2$, we are supposed to prove the below three cases (others, such as $t_2$ being Fix $f$, are trivial just because $t_2$s are not values that yield in contradictions):

             - $t_2 = \lambda x \colon \tau_x. e$, for some type $\tau_x$. Here, we need to show that $\exists t'$, Plus $(\text{NVal } n)$ $(\lambda x \colon \tau_x. e) \to_\beta t'$ holds. This is doable again by contradiction as the term $\lambda x \colon \tau_x. e$ is of an *arrow*-type, while it is expected to be *Int* by the hypothesis ($Hb$).

             - $t_2 = \text{NVal } m$. In this case, we could simply show that $\exists t'$, Plus $(\text{NVal } n)$ $(\text{NVal } m) \to_\beta t'$ holds by first plugging in the term NVal $(n + m)$ for $t'$, and then employing the rule ($plus_1$) presented in Figure 2.

- - $t_2 = $ BVal $b$. The goal $\exists t'$, Plus (NVal $n$) (BVal $b$) $\rightarrow_\beta t'$ in this case holds also by contradiction: the type of (BVal $b$) is *Bool* while it is expected to be *Int* by the rule (*Hb*).

  B. $\exists t'_2$, $t_2 \rightarrow_\beta t'_2$. In this case, $\exists t'$, Plus (NVal $n$) $t_2 \rightarrow_\beta t'$ needs to be shown. In cases where $t_2$ is a value, we close the goal by contradiction, as $\exists t'_2$, $t_2 \rightarrow_\beta t'_2$ allows for proving `False`: values do not evaluate any further. For the rest, we plug the term Plus (NVal $n$) $t'_2$ into the goal, for the term $t'$, and solve it with the rule (*plus$_2$*) stated in Figure 2.

  iii. $t_1 = $ BVal $b$. It is possible to show that $\exists t'$, Plus (BVal $b$) $t_2 \rightarrow_\beta t'$ by contradiction as the type of (BVal $b$) is *Bool* while it is here expected to be *Int* by the rule (*Ha*).

  (b) `isvalue` $t_1 = false$. On a case analysis over the term $t_1$, we in fact have to prove subgoals where $t_1$ is not a value (other cases such as $t_1$ being $\lambda x : \tau. e$ are trivial just because $t_1$s are values that yield in contradictory cases):

  - Recall that we are trying to show $\exists t'$, Plus $t_1$ $t_2 \rightarrow_\beta t'$, such that $t_1$ is not a value. We close all such cases uniformly: there definitely exists a $t'$ as Plus $t'_1$ $t_2$ given $t_1 \rightarrow_\beta t'_1$, thanks to the rule (*plus$_3$*) presented in Figure 2.

5. The remaining cases with, for instance, Minus $t_1$ $t_2$, could be proven by employing a very similar idea presented in the above item 4.

□

We summarize below the given Coq implementation, proof of Theorem 1, in such a way that we highlight the crucial points itemized in the above pen-and-paper proof by comment-outs. Please refer to the accompanying library for the complete proof.

```
Lemma progress: ∀ t T,
  typecheck nil t  = Some T →
  (isvalue t = true) ∨ (∃ t', beta t = Some t').
Proof. intros t.
       induction t; intros T Htc.
       - ...
       - ... (*1*)
         apply istypechecked_app in Htc. (*Lemma 3.1*)
         destruct Htc as (U, (Ha, Hb)).
         specialize (isvalue_dec t1); intros it1. (*1-a*)
         + specialize (IHt2 _ Hb).
           specialize (isvalue_dec t2) as [it2 | it2]. (*1-a-i*)
           ++ case_eq t1; intros.
              * ...
              * ...
                ∃ (subst e x t2). (*1-a-i-bullet_1*)
                ...
              * intros n Ht1. rewrite Ht1 in Htc.
                contradict Htc; easy. (*1-a-i-bullet_2*)
              * intros n Ht1. rewrite Ht1 in Htc.
                contradict Htc; easy. (*1-a-i-bullet_3*)
              * ...
           ++ specialize (IHt2 _ Hb).(*1-a-ii*)
              destruct IHt2 as [it2' | Hstep].
              +++ rewrite it2 in it2'. easy. (*1-a-ii-A*)
              +++ ... (*1-a-ii-B*)
                 case_eq t1.
                 * ...
                 * ... (*1-a-ii-B-bullet_1*)
                   ∃ (App (Lambda x tx e) t2').
                   ...
                 * ...
                   contradict Htc; easy. (*1-a-ii-B-bullet_2*)
                 * ...
                   contradict Htc; easy. (*1-a-ii-B-bullet_3*)
         + specialize (IHt1 _ Ha). (*1-b*)
           destruct IHt1 as [IHt1 | IHt1].
           ++ rewrite it1 in IHt1. contradict IHt1; easy.     (*1-b-i*)
           ++ destruct IHt1 as (t1', Hstep). cbn.
              ...
              case_eq t1; try (intros; ∃ (App t1' t2); easy). (*1-b-ii-bullet*)
```

```
                   ...
      - ... (*2*)
        apply istypechecked_ite in Htc. (*Lemma 3.2*)
        destruct Htc as (Ha, (Hb, Hc)).
        specialize (IHt1 _ Ha).
        specialize (IHt2 _ Hb).
        specialize (IHt3 _ Hc).
        destruct IHt1 as [t1'| Hstep]. (*2-a*)
        + case_eq t1.
          ++ ...
          ++ intros x tx e Ht1. (*2-a-bullet_1*)
             rewrite Ht1 in Ha. contradict Ha. cbn.
             case_eq (typecheck (extend nil x tx) e); easy.
          ++ intros. rewrite H in Ha. contradict Ha; easy. (*2-a-bullet_2*)
          ++ ... (*2-a-bullet_3*)
             case_eq b; intros.
              * ∃ t2. easy.
              * ∃ t3. easy.
        + ... (*2-b*)
          case_eq t1; intros; try (∃ (ITE t1' t2 t3); easy).
          ...
      - apply istypechecked_fix in Htc. (*3*) (*Lemma 3.3*)
        specialize (IHt _ Htc).
        destruct IHt as [ IHt | IHt ].
        + ... (*3-a*)
          case_eq t.
          ++ ...
          ++ ...
             ∃ (subst e x (Fix (Lambda x tx e))). easy (*3-a-bullet_1*)
             ...
          ++ intros n Ht; rewrite Ht in Htc; (*3-a-bullet_2*)
             contradict Htc; easy.
          ++ intros b Ht; rewrite Ht in Htc; (*3-a-bullet_3*)
             contradict Htc; easy.
        + ...
          case_eq t; try (∃ (Fix t'); easy). (*3-b*)
          ...
      - ... (*4*)
        apply istypechecked_plus in Htc. (*Lemma 3.4*)
        destruct Htc as (Ha, (Hb, Hc)).
        specialize (IHt1 _ Ha).
        specialize (IHt2 _ Hb).
        ...
        destruct it1 as [it1 | it1]. (*4-a*)
        + case_eq t1; try (intros; rewrite H in it1; cbn in it1; easy).
           ++ ... (*4-a-i*)
              case_eq (typecheck (extend nil x tx) e); intros;
              rewrite H in Ha; contradict Ha; easy.
           ++ destruct IHt2 as [IHt2 | IHt2]. (*4-a-ii-A*)
              +++ case_eq t2; try (intros; rewrite H in IHt2; cbn in IHt2; easy).
                 * ... (*4-a-ii-A-bullet_1*)
                   case_eq (typecheck (extend nil x tx) e); intros;
                   rewrite H in Hb; contradict Hb; easy.
                 * ... (*4-a-ii-A-bullet_2*)
                   ∃ (NVal (m + n)).
                   ...
                 * ... (*4-a-ii-A-bullet_3*)
                   rewrite Ht2 in Hb.
                   contradict Hb; easy.
              +++ .... (*4-a-ii-B*)
                   case_eq t2; try (∃ (Plus (NVal n) t2'); easy).
                   ...
           ++ ... (*4-a-iii*)
              contradict Ha; easy.
        + ... (*4-b*)
          case_eq t1; try (∃ (Plus t1' t2); easy).
          ...
      - ... (*5*)
Qed.
```

**Lemma 5** (subst_preserves_typing). $\forall x\ t\ v\ \tau\ v\ \Gamma,\ \Gamma, x: v \vdash t: \tau \implies \vdash v: v \implies \Gamma \vdash t[x := v]: \tau$.

**Proof.** The statement informally says that under some context $\Gamma$, substitution of the string $x$ with some term $v$ within the term $t$, the type of $t$ remains unchanged, provided $\Gamma, x: v \vdash t: \tau$ (*Ha*) and $\vdash v: v$ (*Hb*). We argue by *structural induction* on the term $t$:

1.  $t = \text{Ident } s$ for some arbitrary string $s$. If $x = s$, the hypothesis $Ha$ takes the following shape: $\Gamma, s\colon v \vdash \text{Ident } s\colon \tau$ which implies that $v = \tau$. The goal $\Gamma \vdash (\text{Ident } s)[s := v]\colon v$ simplifies into $\Gamma \vdash v\colon v$ by Definition 2 of the substitution function. Employing the fact that $\forall \Gamma\, \Lambda\, e\, \kappa, \Gamma \vdash e\colon \kappa \implies (\forall y, y \notin \text{fv } y \implies \Gamma(y) = \Lambda(y)) \implies \Lambda \vdash e\colon \kappa$ (named `context_invariance` in the Coq code), we deduce $\Gamma \vdash v\colon v$ out of the hypothesis $Hb$, and the goal is closed. Else if $x \neq s$ then $(Ha)$ turns into $\Gamma(s) = \tau$. The goal $\Gamma \vdash (\text{Ident } s)[x := v]\colon \tau$ simplifies into $\Gamma(s) = \tau$ again by Definition 2. This is in fact the hypothesis $(Hb)$ itself.

2.  $t = \lambda s\colon \kappa.\, e$ for some arbitrary string $s$, type $\kappa$, and term $e$. The goal we aim to prove is $\Gamma \vdash (\lambda s\colon \kappa.\, e)[x := v]\colon \tau$, provided an induction hypothesis $\forall x\, v\, \tau\, v\, \Gamma, \Gamma, x\colon v \vdash e\colon \tau \implies \vdash v\colon v \implies \Gamma \vdash e[x := v]\colon \tau\, (IHe)$. If $x = s$ then we have $(\lambda s\colon \kappa.\, e)[s := v]$ amounts to $\lambda s\colon \kappa.\, e$ by Definition 2. Therefore, the goal turns out to be $\Gamma \vdash \lambda s\colon \kappa.\, e\colon \tau$. Using the fact that $\forall \Gamma\, \Lambda\, e\, \kappa, \Gamma \vdash e\colon \kappa \implies (\forall y, y \notin \text{fv } y \implies \Gamma(y) = \Lambda(y)) \implies \Lambda \vdash e\colon \kappa$ (`context_invariance` in the Coq code) over the contexts $\Gamma, s\colon v$ and $\Gamma$ with the hypothesis $Ha$ (namely, $\Gamma, s\colon v \vdash \lambda s\colon \kappa.\, e\colon \tau$), we manage to extend the list of assumptions with $\Gamma \vdash \lambda s\colon \kappa.\, e\colon \tau$, and have the goal closed. If $x \neq s$, we need to show that $\Gamma \vdash \lambda s\colon \kappa.\, e[x := v]\colon \tau$ or better that $\Gamma, s\colon \kappa \vdash e[x := v]\colon \tau$ thanks to Definition 2 and to the $(app_t)$ rule in Figure 4. Remark that in this case, the hypothesis $(Ha)$ first takes the shape $\Gamma, x\colon v \vdash \lambda s\colon \kappa.\, e\colon \tau$ then simplifies into $\Gamma, x\colon v, s\colon \kappa \vdash e\colon \tau$ again by the $(app_t)$ rule. Specializing the induction hypothesis $(IHe)$ with the context $\Gamma, s\colon \kappa$, simplified version of $(Ha)$ and $(Hb)$, we close this goal.

3.  $t = (t_1\ t_2)$ for some terms $t_1$ and $t_2$. We try proving $\Gamma \vdash (t_1\ t_2)[x := v]\colon \tau$ given $\forall x\, v\, \tau\, v\, \Gamma, \Gamma, x\colon v \vdash t_1\colon \tau \implies \vdash v\colon v \implies \Gamma \vdash t_1[x := v]\colon \tau\, (IHt_1)$ and $\forall x\, v\, \tau\, v\, \Gamma, \Gamma, x\colon v \vdash t_2\colon \tau \implies \vdash v\colon v \implies \Gamma \vdash t_2[x := v]\colon \tau\, (IHt_2)$ as induction hypotheses. We deduce $\Gamma, x\colon v \vdash t_1\colon \kappa \to \tau\, (H)$ and $\Gamma, x\colon v \vdash t_2\colon \kappa\, (H_0)$, for some type $\kappa$, by inversion over the hypothesis $(Ha)$, namely $\Gamma, x\colon v \vdash (t_1\ t_2)\colon \tau$. Using $(H)$ and $(Hb)$ in $(IHt_1)$, and $(H_0)$ and $(Hb)$ in $(IHt_2)$, we, respectively, obtain $\Gamma \vdash t_1[x := v]\colon \kappa \to \tau$ and $\Gamma \vdash t_2[x := v]\colon \kappa$, which prove the goal thanks to Definition 4 of the substitution function and the rule $(app_t)$ in Figure 4.

4.  $t = \text{ITE } t_1\ t_2\ t_3$ for some terms $t_1$, $t_2$, and $t_3$. The statement we intend to show in this case turns out to be $\Gamma \vdash (\text{ITE } t_1\ t_2\ t_3)[x := v]\colon \tau$ given $\forall x\, v\, \tau\, v\, \Gamma, \Gamma, x\colon v \vdash t_1\colon \tau \implies \vdash v\colon v \implies \Gamma \vdash t_1[x := v]\colon \tau\, (IHt_1)$, $\forall x\, v\, \tau\, v\, \Gamma, \Gamma, x\colon v \vdash t_2\colon \tau \implies \vdash v\colon v \implies \Gamma \vdash t_2[x := v]\colon \tau\, (IHt_2)$ and $\forall x\, v\, \tau\, v\, \Gamma, \Gamma, x\colon v \vdash t_3\colon \tau \implies \vdash v\colon v \implies \Gamma \vdash t_3[x := v]\colon \tau\, (IHt_3)$ as induction hypotheses. It is quite easy to infer $\Gamma, x\colon v \vdash t_1\colon Bool\, (H)$, $\Gamma, x\colon v \vdash t_2\colon \tau\, (H_0)$, and $\Gamma, x\colon v \vdash t_3\colon \tau\, (H_1)$ by inversion over the hypothesis $(Ha)$, namely $\Gamma, x\colon v \vdash \text{ITE } t_1\ t_2\ t_3\colon \tau$. We then specialize $(IHt_1)$ with $(H)$ and $(Hb)$, $(IHt_2)$ with $(H_0)$ and $(Hb)$, and $(IHt_3)$ with $(H_1)$ and $(Hb)$ to respectively obtain $\Gamma \vdash t_1[x := v]\colon Bool$, $\Gamma \vdash t_2[x := v]\colon \tau$ and $\Gamma \vdash t_3[x := v]\colon \tau$. These are adequate to prove the statement due to Definition 4 of the substitution function and the rule $(ite_t)$ in Figure 4.

5.  $t = \text{Fix } t_1$ for some term $t_1$. The goal of the case is of the following shape: $\Gamma \vdash (\text{Fix } t_1)[x := v]\colon \tau$. We additionally have a single induction hypothesis $\forall x\, v\, \tau\, v\, \Gamma, \Gamma, x\colon v \vdash t_1\colon \tau \implies \vdash v\colon v \implies \Gamma \vdash t_1[x := v]\colon \tau\, (IHt_1)$. The hypothesis $(Ha)$, $\Gamma, x\colon v \vdash \text{Fix } t_1\colon \tau$, entails by inversion that $\Gamma, x\colon v \vdash t_1\colon \tau \to \tau\, (H)$. We then specialize $(IHt_1)$ with $(H)$ and $(Hb)$ to have $\Gamma \vdash t_1[x := v]\colon \tau \to \tau$. This is enough to close the goal thanks to Definition 4 of the substitution function and the rule $(fix_t)$ in Figure 4.

6.  $t = \text{Plus } t_1\ t_2$ for some terms $t_1$ and $t_2$. The goal we aim to close in this case is $\Gamma \vdash (\text{Plus } t_1\ t_2)[x := v]\colon \tau$ along with two induction hypotheses $\forall x\, v\, \tau\, v\, \Gamma, \Gamma, x\colon v \vdash t_1\colon \tau \implies \vdash v\colon v \implies \Gamma \vdash t_1[x := v]\colon \tau\, (IHt_1)$, $\forall x\, v\, \tau\, v\, \Gamma, \Gamma, x\colon v \vdash t_2\colon \tau \implies \vdash v\colon v \implies \Gamma \vdash t_2[x := v]\colon \tau\, (IHt_2)$. We infer $\Gamma, x\colon v \vdash t_1\colon Int\, (H)$ and $\Gamma, x\colon v \vdash t_2\colon Int\, (H_0)$ inverting the hypothesis $(Ha)$. Lastly, we specialize $(IHt_1)$ with $(H)$ and $(Hb)$, $(IHt_2)$ with $(H_0)$ and $(Hb)$ to handle $\Gamma \vdash t_1[x := v]\colon Int$, $\Gamma \vdash t_2[x := v]\colon Int$ which prove the goal due to Definition 4 and the rule $(plus_t)$ in Figure 4.

7.  The cases with $t$, being either Minus $t_1$ $t_2$, Mult $t_1$ $t_2$, Eq $t_1$ $t_2$, or Gt $t_1$ $t_2$ follow the same lines with the proof given in the above item 6. The cases where $t = $ NVal $n$ and $t = $ BVal $b$ are trivial just because the substitution function has no impact on these terms.

□

The preservation statement claims that the type of a given term does not vary under beta-reduction.

**Theorem 2** (Preservation). $\forall t\ t',\ \vdash t\colon \tau\ \wedge\ t \to_\beta t'\ \implies\ \vdash t'\colon \tau.$

**Proof.** The proof proceeds by a structural induction over the term $t$. The cases involving Ident $x$, $\lambda x\colon \tau.\ e$, NVal $n$, and BVal $b$ trivially hold by contradiction: these terms do not reduce any further, contradicting the assumption $t \to_\beta t'$.

1.  The case with the application $(t_1\ t_2)$, for some arbitrary terms $t_1$ and $t_2$, are more appealing and, thus, deserve a closer look. Here, we are supposed to show (for all types $\tau$) that $\vdash t'\colon \tau$ holds; provided $(t_1\ t_2)\colon \tau$ $(H)$, $(t_1\ t_2) \to_\beta t'$ $(H_0)$, and a pair of induction hypotheses $\forall t'\ \tau,\ \vdash t_1\colon \tau\ \wedge\ t_1 \to_\beta t'\ \implies\ \vdash\ t'\colon \tau$ $(IHt_1)$ and $\forall t'\ \tau,\ \vdash t_2\colon \tau\ \wedge\ t_2 \to_\beta t'\ \implies\ \vdash\ t'\colon \tau$ $(IHt_2)$. Notice that by plugging in the hypothesis $(H)$ into Lemma 1, we can deduce the facts that $\vdash t_1\colon v \to \tau$ $(H_1)$ and $\vdash t_2\colon v$ $(H_2)$, for some type $v$. At this stage, we apply a case analysis on the term $t_1$ (below the goals) to close:

    (a)  $t_1 = \lambda x\colon v.\ e$ for some term $e$ and type $v$. We definitely obtain some $t'$ after beta-reducing the term inhabited by the hypothesis $(H)$ $(\lambda x\colon v.\ e)\ t_2$ depending on the choice of whether $t_2$ is a value or not:

        i.  `isvalue` $t_2 = \mathit{true}$. In this case, $t'$ amounts to $e[x := t_2]$, due to the rule $(app_3)$ presented in Figure 2, and we are expected to prove that $\vdash e[x := t_2]\colon \tau$. Thanks to Lemma 5, to obtain $\vdash e[x := t_2]\colon \tau$ (proven), we need to close two goals, which are $x\colon v \vdash e\colon \tau$ and $\vdash t_2\colon v$:

            •  $x\colon v \vdash e\colon \tau$. Recall that we have $\vdash \lambda x\colon v.\ e\colon v \to \tau$ due to $(H_1)$. We solve the goal just by inverting the rule $(lam_t)$ in Figure 4.
            •  $\vdash t_2\colon v$. This one is exactly $(H_2)$.

        ii.  `isvalue` $t_2 = \mathit{false}$. Thanks to Progress Theorem 1 and the hypothesis $(H_1)$, we have `isvalue` $t_2 = \mathit{true} \vee \exists t_2',\ t_2 \to_\beta t_2'$. As the left side of the disjunction is contradictory to the assumption of the case, we focus on the right side, which tells us that there exists some $t_2'$, such that $t_2 \to_\beta t_2'$. Therefore, $t'$ amounts to $(\lambda x\colon v.\ e)\ t_2'$ due to the rule $(app_4)$ presented in Figure 2. Namely, we are supposed to show that $\vdash (\lambda x\colon v.\ e)\ t_2'\colon \tau$ holds. By properly specializing the induction hypothesis $(IHt_2)$, we end up with $\vdash t_2'\colon v$. With this information in hand, just by employing the rule $(app_t)$ in Figure 4, we ensure that the application $(\lambda x\colon v.\ e)\ t_2'$ is of type $\tau$ under the empty context.

    (b)  $t_1 = (e_1\ e_2)$ for some arbitrary terms $e_1$ and $e_2$. It is known by $(H_1)$ that the application $(e_1\ e_2)$ is of type $v \to \tau$ under the empty context. Passing this well-typed information to Progress Theorem 1, we have `isvalue` $(e_1\ e_2) = \mathit{true} \vee \exists t_1',\ (e_1\ e_2) \to_\beta t_1'$. Due to the fact that `isvalue` $(e_1\ e_2) = \mathit{false}$, we are left with $\exists t_1',\ (e_1\ e_2) \to_\beta t_1'$. Therefore, the goal that we aim to prove in this case takes the following shape: $\vdash t_1'\ t_2\colon \tau$ due to the rule $(app_2)$ presented in Figure 2. Specializing the induction hypothesis $(IHt_1)$ with the correct ingredients gives us $\vdash t_1'\colon v \to \tau$. Making use of the hypothesis $(H_2)$ and the rule $(app_t)$ placed in Figure 4, we conclude that $\vdash t_1'\ t_2\colon \tau$ holds.

    (c)  $t_1 = $ ITE $e_1\ e_2\ e_3$ for some arbitrary terms $e_1$, $e_2$ and $e_3$. Similar to the proof in the above item, we know that the term ITE $e_1\ e_2\ e_3$ is of type $v \to \tau$ under

the empty context. This, Progress Theorem 1 gives us isvalue ITE $e_1$ $e_2$ $e_3$ = *true* $\vee$ $\exists t'_1$, ITE $e_1$ $e_2$ $e_3$ $\rightarrow_\beta$ $t'_1$. Just that isvalue ITE $e_1$ $e_2$ $e_3$ = *true* is incorrect, we focus on $\exists t'_1$, (ITE $e_1$ $e_2$ $e_3$) $\rightarrow_\beta$ $t'_1$. This heads us toward proving $\vdash t'_1$ $t_2$ : $\tau$ due to the rule ($app_2$) presented in Figure 2. Similar to that of the above item (b), we specialize in the induction hypothesis ($IHt_1$) with the correct ingredients, and have $\vdash t'_1$ : $v \rightarrow \tau$. We close this goal just by employing the hypothesis ($H_2$) and the rule ($app_t$) appearing in Figure 4.

(d)    $t_1$ = Fix $e_1$ for some arbitrary term $e_1$. By chasing the exact same steps demonstrated in the above items (b) and (c), we end up retaining $\exists t'_1$, Fix $e_1$ $\rightarrow_\beta$ $t'_1$ to show $\vdash t'_1$ $t_2$ : $\tau$, which we solve again by putting the induction hypothesis ($IHt_2$) together with the rule ($app_t$) in operation.

(e)    $t_1$ = (Plus $e_1$ $e_2$) for some arbitrary terms $e_1$ and $e_2$. The goal in this case is proven by contradiction as due to the hypothesis ($H_1$), the term $t_1$ needs to be of some arrow-type $v \rightarrow \tau$, but it is of type *Int*.

(f)    The other cases in which the term $t_1$ appears to be either NVal $n$, Minus $e_1$ $e_2$, Mult $e_1$ $e_2$, Eq $e_1$ $e_2$, or Gt $e_1$ $e_2$, and could be proven in a similar manner with that of the above item (e). The goal where $t_1$ amounts to BVal $b$ is similarly proven with a single difference, where $t_1$ is of type *Bool*, not *Int*. Lastly, the goal with $t_1$ = Ident $x$ holds by contradiction as the term $t_1$ is ill-typed under the empty context.

2.    $t$ = ITE $t_1$ $t_2$ $t_3$ for some terms $t_1$, $t_2$ and $t_3$. In this case, we aim to prove for all types $\tau$ that $\vdash t'$ : $\tau$ holds; given that ITE $t_1$ $t_2$ $t_3$ : $\tau$ ($H$), ITE $t_1$ $t_2$ $t_3$ $\rightarrow_\beta$ $t'$ ($H_0$) along with three induction hypotheses $\forall t'$ $\tau$, $\vdash t_1$ : $\tau$ $\wedge$ $t_1$ $\rightarrow_\beta$ $t'$ $\implies \vdash t'$ : $\tau$ ($IHt_1$), $\forall t'$ $\tau$, $\vdash t_2$ : $\tau$ $\wedge$ $t_2$ $\rightarrow_\beta$ $t'$ $\implies \vdash t'$ : $\tau$ ($IHt_2$), and $\forall t'$ $\tau$, $\vdash t_3$ : $\tau$ $\wedge$ $t_3$ $\rightarrow_\beta$ $t'$ $\implies \vdash t'$ : $\tau$ ($IHt_3$). Note also that we deduce the facts $\vdash t_1$ : *Bool* ($Ha$), $\vdash t_2$ : $\tau$ ($Hb$), and $\vdash t_3$ : $\tau$ ($Hc$) by specializing Lemma 2 with the hypothesis ($H$). The proof proceeds with a case analysis over the term $t_1$, and requires the following cases to be proven:

(a)    The goals in which the term $t_1$ is Ident $x$, NVal $n$, $\lambda x$ : $v$. $e$, Plus $e_1$ $e_2$, Minus $e_1$ $e_2$, Mult $e_1$ $e_2$, Eq $e_1$ $e_2$, or Gt $e_1$ $e_2$ are trivially shown by contradictions, as none of these terms are of type *Bool* as expected by the hypothesis ($Ha$).

(b)    $t_1$ = ($e_1$ $e_2$) for some arbitrary terms $e_1$ and $e_2$. The hypothesis $Ha$ tells us that $\vdash (e_1$ $e_2)$ : *Bool*. This, with Progress Theorem 1, entails that isvalue ($e_1$ $e_2$) = *true* $\vee$ $\exists t'_1$, ($e_1$ $e_2$) $\rightarrow_\beta$ $t'_1$. The left side of the disjunction is obviously incorrect. We, therefore, obtain $\exists t'_1$, ($e_1$ $e_2$) $\rightarrow_\beta$ $t'_1$. Accordingly, the goal we intend to prove in this case turns out to be $\vdash$ ITE $t'_1$ $t_2$ $t_3$ : $\tau$ due to the rule ($ite_3$) presented in Figure 2. By specializing the induction hypothesis ($IHt_1$) with proper terms and types, we have $\vdash t'_1$ : *Bool*. Thanks to the hypotheses ($Hb$), ($Hc$), and the rule ($ite_t$) given in Figure 4, we show that $\vdash$ (ITE $t'_1$ $t_2$ $t_3$) : $\tau$.

(c)    $t_1$ = BVal $b$ for Boolean $b$. We carry on with a case distinction on $b$:

- if $b$ is Boolean *true*: ITE (BVal true) $t_2$ $t_3$ $\rightarrow_\beta$ $t_2$, thanks to the rule ($ite_1$) presented in Figure 2. Therefore, the goal takes the shape $\vdash t_2$ : $\tau$, which is the hypothesis ($Hb$).

- if $b$ is Boolean *false*: similarly, ITE (BVal false) $t_2$ $t_3$ $\rightarrow_\beta$ $t_3$, thanks to the rule ($ite_2$) stated in Figure 2. The goal is now $\vdash t_3$ : $\tau$. This is trivial as it is exactly the hypothesis ($Hc$).

(d)    $t_1$ = ITE $e_1$ $e_2$ $e_3$ for some arbitrary terms $e_1$, $e_2$ and $e_3$. The hypothesis ($Ha$) entails that $\vdash$ ITE $e_1$ $e_2$ $e_3$ : *Bool*. Progress Theorem 1 over this fact gives isvalue ITE $e_1$ $e_2$ $e_3$ = *true* $\vee$ $\exists t'_1$, ITE $e_1$ $e_2$ $e_3$ $\rightarrow_\beta$ $t'_1$. Provided that isvalue ITE $e_1$ $e_2$ $e_3$ = *false*, we focus on the right side of the disjunction; that is $\exists t'_1$, ITE $e_1$ $e_2$ $e_3$ $\rightarrow_\beta$ $t'_1$. In this parallel, the goal we want to close is $\vdash$ ITE $t'_1$ $t_2$ $t_3$ : $\tau$ thanks to the rule ($ite_3$) presented in Figure 2. By specializing the induction hypothesis ($IHt_1$) with proper terms and types, we have

⊢ $t_1'$: *Bool*. Thanks to the hypotheses (*Hb*), (*Hc*), and the rule (*ite_t*) stated in Figure 4, we show that ⊢ ITE $t_1'$ $t_2$ $t_3$: $\tau$.

(e)　$t_1 =$ Fix $f$ for some term $f$. Similar to case (d) above, the hypothesis (*Ha*) entails that ⊢ Fix $f$: $\tau$. This, with Progress Theorem 1, we deduce `isvalue` Fix $f =$ *true* $\lor$ $\exists f'$, Fix $f \to_\beta f'$. As it is obvious that `isvalue` Fix $f =$ *false*, we are left with $\exists f'$, Fix $f \to_\beta f'$. The goal we want to close here is that ⊢ ITE $f'$ $t_2$ $t_3$: $\tau$ due to the rule (*ite_3*) presented in Figure 2. We now specialize the induction hypothesis (*IHt_1*) with proper terms and types, and obtain ⊢ $f'$: *Bool*. Thanks to the hypotheses (*Hb*), (*Hc*), and the rule (*ite_t*) appearing in Figure 4, we prove that ⊢ ITE $f'$ $t_2$ $t_3$: $\tau$.

3.　For the case Fix $f$, we aim to prove for all types $\tau$ that ⊢ $f'$: $\tau$ holds; given that Fix $f$: $\tau$ (*H*), Fix $f \to_\beta t'$ (*H_0*) strengthened with the induction hypothesis $\forall f' \tau$, ⊢ $f$: $\tau$ $\land$ $f \to_\beta f'$ $\implies$ ⊢ $f'$: $\tau$ (*IHf*). Moreover, we have ⊢ $f$: $\tau \to \tau$ (*Ha*) when Lemma 3 is applied to the hypothesis (*H*). The proof proceeds with a case analysis over the term $f$, and requires the following cases to be proven:

(a)　The goals in which the term $t_1$ is any of Ident $x$, NVal $n$, BVal $b$, Plus $e_1$ $e_2$, Minus $e_1$ $e_2$, Mult $e_1$ $e_2$, Eq $e_1$ $e_2$ and Gt $e_1$ $e_2$ are trivially demonstrated by contradiction as none of these terms are of the arrow type $\tau \to \tau$ as expected by the hypothesis (*Ha*).

(b)　$f = \lambda x$: $v.\, e$ for some term $e$ and type $v$. Recall that the rule (*fix_1*) stated in Figure 2, we have Fix ($\lambda x$: $v.\, e$) $\to_\beta e[x :=$ Fix ($\lambda x$: $v.\, e$)]. Notice that, on a side note, the term $\lambda x$: $v.\, e$ needs to be of type $\tau \to \tau$ due to the hypothesis (*Ha*). By inversion here, we deduce that $v = \tau$ and $x$: $\tau \vdash e$: $\tau$ (*Hb*). Having said that, let us look into the statement that needs to be proven in this case: ⊢ $e[x :=$ Fix ($\lambda x$: $\tau.\, e$)]: $\tau$ (due to the rule (*fix_1*) presented in Figure 2). Thanks to Lemma 5, to prove the mentioned goal, we need to close two goals: $x$: $\tau \vdash e$: $\tau$ and ⊢ Fix ($\lambda x$: $\tau.\, e$): $\tau$.

- $x$: $\tau \vdash e$: $\tau$. This is exactly the hypothesis (*Hb*).
- ⊢ Fix ($\lambda x$: $\tau.\, e$): $\tau$. This one matches with the hypothesis (*H*).

(c)　$f = (e_1\, e_2)$ for some arbitrary terms $e_1$ and $e_2$. The hypothesis (*Ha*) entails that ⊢ $(e_1\, e_2)$: $\tau \to \tau$. Employing this fact within Progress Theorem 1, we deduce `isvalue` $(e_1\, e_2) =$ *true* $\lor$ $\exists f'$, $(e_1\, e_2) \to_\beta f'$. The left side of the disjunction is obviously incorrect. We therefore obtain $\exists f'$, $(e_1\, e_2) \to_\beta f'$. Thus the goal turns out to be ⊢ Fix $f'$: $\tau$ due to the rule (*fix_2*) presented in Figure 2. By properly specializing the induction hypothesis (*IHf*), we have ⊢ $f'$: $\tau \to \tau$. Now by the rule (*fix_t*) given in Figure 4, we conclude that Fix $f'$: $\tau$.

(d)　$f =$ ITE $e_1$ $e_2$ $e_3$ for some arbitrary terms $e_1$, $e_2$ and $e_3$. Similar to case (c) above, the hypothesis (*Ha*) entails that ⊢ ITE $e_1$ $e_2$ $e_3$: $\tau \to \tau$. Progress Theorem 1 specialized with this fact implies that `isvalue` ITE $e_1$ $e_2$ $e_3 =$ *true* $\lor$ $\exists t_1'$, ITE $e_1$ $e_2$ $e_3 \to_\beta t_1'$. Provided that `isvalue` ITE $e_1$ $e_2$ $e_3 =$ *false*, we focus on the right side of the disjunction; that is $\exists f'$, ITE $e_1$ $e_2$ $e_3 \to_\beta f'$. In this parallel, the goal we want to close here is ⊢ Fix $f'$: $\tau$ due to the rule (*fix_2*) presented in Figure 2. We now specialize the induction hypothesis (*IHf*), and obtain ⊢ $f'$: $\tau \to \tau$. Now by the rule (*fix_t*) stated in Figure 4, we conclude that Fix $f'$: $\tau$.

(e)　$f =$ Fix $e$ for some arbitrary term $e$. By following the exact same steps presented in the above items (c) and (d), we end up having $\exists f'$, Fix $e \to_\beta f'$ to show ⊢ Fix $f'$: $\tau$ (due to the rule (*fix_2*) presented in Figure 2) which we solve again by putting the induction hypothesis (*IHf*) together with the rule (*fix_t*) in use.

4.　Concerning the case with $t =$ Plus $t_1$ $t_2$, for some arbitrarily chosen terms, $t_1$ and $t_2$, we want to prove for all types $\tau$ that ⊢ $t'$: $\tau$ holds, provided that ⊢ Plus $t_1$ $t_2$: $\tau$ (*H*), Plus $t_1$ $t_2 \to_\beta t'$ (*H_0*) and two induction hypotheses $\forall t' \tau$, ⊢ $t_1$: $\tau$ $\land$ $t_1 \to_\beta t'$ $\implies$

$\vdash t': \tau$ ($IHt_1$), $\forall t'\ \tau$, $\vdash t_2: \tau \land t_2 \rightarrow_\beta t' \implies \vdash t': \tau$ ($IHt_2$). In addition to these, we enrich the set of hypotheses with $\tau = Int$ ($Ha$), $\vdash t_1: Int$ ($Hb$), and $\vdash t_2: Int$ ($Hc$) just by applying Lemma 4 over the hypothesis ($H$). The proof proceeds with a case distinction over the term $t_1$, and throws us cases to prove:

(a)     The goals in which term $t_1$ is Ident $x$, BVal $b$, ($\lambda x: v.e$), (Eq $e_1$ $e_2$), or (Gt $e_1$ $e_2$) and trivially proven by contradiction as none of these terms are of type *Int* as expected by the hypothesis ($Hb$).

(b)     $t_1 = (e_1\ e_2)$ for some arbitrary terms $e_1$ and $e_2$. We deduce out of the hypothesis ($Hb$) that $\vdash (e_1\ e_2): Int$. With this information, we could further infer that `isvalue` $(e_1\ e_2) = true \lor \exists t_1', (e_1\ e_2) \rightarrow_\beta t_1'$ thanks to Progress Theorem 1. Given that `isvalue` $(e_1\ e_2) = false$, we take $\exists t_1', (e_1\ e_2) \rightarrow_\beta t_1'$ into account, and therefore the goal takes the following shape: $\vdash$ Plus $t_1'\ t_2: Int$, due to the rule ($plus_3$) presented in Figure 2. We could put the induction hypothesis ($IHt_1$) into the following shape: $\vdash t_1': Int$ employing the hypothesis ($Hb$). Now, by using the rule ($plus_t$), and the hypothesis ($Hc$), we conclude that $\vdash$ Plus $t_1'\ t_2: Int$ holds.

(c)     $t_1 = $ NVal $n$ for some Coq natural $n$. Thanks to the hypothesis ($Hc$), we already know that $\vdash t_2: Int$. Applying Progress Theorem 1 to this fact, we obtain `isvalue` $t_2 = true \lor \exists t_2', t_2 \rightarrow_\beta t_2'$. Destructing this disjunction throws us the following two goals to prove, independently assuming `isvalue` $t_2 = true$ and $\exists t_2', t_2 \rightarrow_\beta t_2'$:

- `isvalue` $t_2 = true$. The only choice that makes this case non-contradictory is that of $t_2 = $ NVal $m$ for a Coq natural $m$. Other cases lead to contradictions disobeying either `isvalue` $t_2 = true$ or $\vdash t_2: Int$. Using the rule ($plus_1$) shown in Figure 2, we have Plus (NVal $n$) (NVal $m$) $\rightarrow_\beta$ NVal $(n+m)$. Therefore, the statement we try proving here is $\vdash$ NVal $(n+m): Int$, which is entailed by the rule ($nval_t$) presented in Figure 4.

- $\exists t_2', t_2 \rightarrow_\beta t_2'$. We proceed with a case distinction on the term $t_2$. Note that the statements where $t_2$ is a value trivially hold, no further reductions from $t_2$ are possible, which contradicts the assumption of the case. We have the statement $\vdash$ Plus (NVal $n$) $t_2': Int$ to be proven for the remaining cases, due to the rule ($plus_2$) stated in Figure 2. To prove this statement, we specialize the induction hypothesis ($IHt_2$) with proper terms and types, and turn it into $\vdash t_2': Int$. This fact and the rule ($plus_t$) presented in Figure 4 give us $\vdash$ Plus (NVal $n$) $t_2': Int$.

(d)     $t_1 = $ ITE $e_1\ e_2\ e_3$ for some arbitrary terms $e_1$, $e_2$ and $e_3$. The hypothesis ($Hb$) can be used to infer $\vdash$ ITE $e_1\ e_2\ e_3: Int$. Using this within Progress Theorem 1, it is possible to infer `isvalue` ITE $e_1\ e_2\ e_3 = true \lor \exists t_1',$ ITE $e_1\ e_2\ e_3 \rightarrow_\beta t_1'$. Recall that only the right side of this disjunction is useful as the other leads to a contradiction. Building on this, our goal here turns out to be $\vdash$ Plus $t_1'\ t_2: Int$ due to the rule ($plus_3$) given in Figure 2. We then specialize the induction hypothesis ($IHt_1$) properly and obtain $\vdash t_1': Int$. By the rule ($plus_t$) and the hypothesis ($Hc$), we have $\vdash$ Plus $t_1'\ t_2: Int$ proven.

(e)     $t_1 = $ Fix $f$ for some arbitrary term $f$. We follow the exact same steps presented in the above items (b) and (d): first infer $\exists t_1',$ Fix $f \rightarrow_\beta t_1'$ then show $\vdash$ Plus $t_1'\ t_2: Int$ by employing the rule ($plus_t$), and putting the induction hypothesis ($IHt_1$) in the intended shape.

(f)     $t_1 = $ Plus $e_1\ e_2$ for some arbitrary terms $e_1$ and $e_2$. We follow the same step with that of the above item (e). That is, we first have $\exists t_1',$ Plus $e_1\ e_2 \rightarrow_\beta t_1'$ out of Progress Theorem 1, and aim at proving $\vdash$ Plus $t_1'\ t_2: Int$ (thanks to the rule ($plus_3$) presented in Figure 2). The proof is constructed out of the rule ($plus_t$) and correctly shaped induction hypothesis ($IHt_1$).

(g)     The other cases in which the term $t_1$ appears to be Minus $e_1$ $e_2$ or Mult $e_1$ $e_2$ could be proven in a similar manner to Plus $e_1$ $e_2$ described in the above item (f).

5.     The remaining cases with, for instance, Mult $t_1$ $t_2$, could be proven, employing a very similar idea presented in the above item 4.

□

We summarize below, in a Coq implementation, a proof of Theorem 2 marking the items presented in the above pen-and-paper proof with comment-outs. Please refer to the accompanying library for the complete proof.

```
Lemma preservation: ∀ (t t': term) (T: type),
   typecheck nil t = Some T ∧ beta t = Some t' → typecheck nil t' = Some T.
Proof. intro t.
      induction t; intros t' T (H, H0).
      ...
      - ... (*1*)
        apply istypechecked_app in H. (*Lemma 3.1*)
        destruct H as (U, (H1, H2)).
        case_eq t1.
        + ...
        + intros x v e Ht1. (*1-a: t₁ = λx: v. e*)
          ...
          case_eq (isvalue t2); intros.
          ++ ... (*1-a-i*)
             specialize (subst_preserves_typing e t2 x T U nil); intros. (*Lemma 3.5*)
             +++ ... (*1-a-i-bullet_1*)
                 case_eq (typecheck (extend nil x v) e); intros.
                 * ...
                 * rewrite H6 in H1. contradict H1; easy.
             +++ exact H2. (*1-a-bullet_2*)
          ++ ... (*1-a-ii*)
             specialize (progress t2 U H2); intros. (*Theorem 3.1*)
             destruct H3 as [ H3 | H3 ].
             +++ ...
                 contradict H3; easy.
             +++ ...
                 specialize (IHt2 t2' U (conj H2 H3)).
                 rewrite IHt2, type_eqb_refl. easy.
        + intros e1 e2 Ht1. (*1-b: t₁ = e₁ e₂*)
          ...
          assert (isvalue (App e1 e2) = false) by easy.
          ...
          specialize (progress t1 (Arrow U T) H1); intros. (*Theorem 3.1*)
          destruct H3 as [ H3 | H3 ].
          ++ ... contradict H3; easy.
          ++ ...
             specialize (IHt1 t1' (Arrow U T) (conj H1 H3)).
             rewrite IHt1, H2, type_eqb_refl. easy.
        + ...
        + intros e1 e2 Ht1. (*1-e: t₁ = Plus e₁ e₂*)
          ...
          case_eq (typecheck nil e1); intros.
          ++ ...
             case_eq (typecheck nil e2); intros.
             +++ ... destruct t; contradict H1; easy.
             +++ ... contradict H0. easy.
             +++ contradict H. easy.
             +++ contradict H. easy.
        + ...
      - ... (*2*)
        apply istypechecked_ite in H. (*Lemma 3.2*)
        destruct H as (Ha, (Hb, Hc)).
        case_eq t1.
        + ...
        + intros e1 e2 Ht1. (*2-b: t₁ = e₁ e₂*)
          ...
          assert (isvalue (ITE t1 t2 t3) = false) by easy.
          ...
          specialize (progress (App e1 e2) Bool Ha ); intros. (*Theorem 3.1*)
          destruct H1 as [H1 | H1].
          ++ ... contradict H1; easy.
          ++ destruct H1 as (t1', H1).
             ...
             specialize (IHt1 t1' Bool (conj Ha H1)).
```

```
              rewrite IHt1, Hb, Hc, !type_eqb_refl. easy.
        + intros b Ht1. (*2-c: t₁ = BVal b*)
          rewrite Ht1 in H0.
          case_eq b; intros.
          ++ ... rewrite ← H2. easy. (*2-c-bullet_1*)
          ++ ... rewrite ← H2. easy. (*2-c-bullet_2*)
        + ...
      - ... (*3*)
        apply istypechecked_fix in H. (*Lemma 3.3*)
        case_eq t.
        + ...
        + intros x v e Ht1. (*3-b: t₁ = λx: v. e*)
          ...
          case_eq (typecheck (extend nil x v) e); intros.
          ++ rewrite H1 in H.
            specialize (subst_preserves_typing
                           e (Fix (Lambda x v e)) x T T nil); intros. (*Lemma 3.5*)
            ...
            +++ ...
            +++ inversion H. subst. easy.
            +++ rewrite Ht1 in Ha. easy.
          ++ ...
      - apply istypechecked_plus in H. (*4*) (*Lemma 3.4*)
        destruct H as (Ha, (Hb, Hc)).
        ...
        case_eq t1.
        + ...
        + intros n Ht1. (*4-c: t₁ = NVal n*)
          rewrite Ht1 in H0. cbn in H0.
          specialize (progress t2 Int Hb); intros. (*Theorem 3.1*)
          destruct H as [H | H].
          ++ ... (*4-c-bullet_1*)
            case_eq t2; try (intros; rewrite H1 in H0; contradict H; easy).
            ...
            rewrite Ht2 in H0. inversion H0. cbn. subst. easy.
          ++ destruct H as (t2', H). (*4-c-bullet_2*)
            rewrite H in H0.
            case_eq t2.
            +++ ...
            +++ ...
                specialize (IHt2 t2' Int (conj Hb H)).
                rewrite IHt2, Hc. easy.
      - ... (*5*)
      - ...
  Qed.
```

**Definition 5** (Multi-Step Evaluation). *We define the multi-step evaluation function (evaln) as the reflexive-transitive closure of the beta-reduction upper-bounded by a natural n:*

$$
\begin{aligned}
\text{evaln } t \; 0 &\quad \rightarrow \quad t \\
\text{evaln } t \; (S \; n) &\quad \rightarrow \quad \text{let } t \rightarrow_\beta t' \text{ in } \text{evaln } t' \; n
\end{aligned}
$$

*In Coq, we wrap the output term of the evaln function with Coq's* option *type. By this, we aim to handle the cases in which t has no further evaluation steps with* None:

```
Fixpoint evaln (t: term) (n: nat): option term ≜
  match n with
    | 0   ⇒ Some t
    | S n ⇒ let t' ≜ beta t in
              match t' with
                | Some t' ⇒ evaln t' n
                | None    ⇒ None
              end
  end.
```

**Definition 6** (Stuck). *A term t is said to be* stuck *if there exists no t', such that t $\rightarrow_\beta$ t' and t is not a value. That formally is*

$$
\forall t, \; \nexists t', \; t \rightarrow_\beta t' \; \wedge \; \text{isvalue } t = \text{false}.
$$

It is straightforward to reflect this definition into Coq:

```
Definition stuck (t: term)≜ beta t = None ∧ isvalue t = false.
```

The soundness statement just combines the claims of that of progress and preservation. Namely, for every well-typed term $t$, if after $n$ reduction steps, $t$ reduces into some term $t'$ then $t'$ cannot be stuck.

**Theorem 3** (Type Soundness). $\forall n\ t\ t'\ \tau, \vdash t: \tau \ \wedge\ evaln\ t\ n = t' \implies \neg stuck\ t'.$

**Proof.** Unfolding the definitions `stuck` and `not`, we are supposed to prove `False` (in Coq's Prop) provided that $\vdash t: \tau$ (*Ha*), $evaln\ t\ n = t'$ (*Hb*), $\nexists t'', t' \to_\beta t''$ (*Hne*) and $isvalue\ t' = false$ (*Hnv*). The proof proceeds by a structural induction over the natural number $n$ throwing us two cases to prove.

1.  $n = 0$. Notice that the hypothesis (*Hb*) (with $n = 0$) entails that $t = t'$. We employ Progress Theorem 1 specialized by the hypothesis (*Ha*), and deduce $isvalue\ t = true \vee \exists t'', t \to_\beta t''$. The goal, in this case, is trivially proven as the left side of the disjunction contradicts with (*Hnv*), while the right-hand side contradicts with (*Hne*).

2.  $n = S\ m$. In this case, we additionally have the induction hypothesis $\forall t\ t'\ \tau, \vdash t: \tau \implies evaln\ t\ m = t' \implies \nexists t'',\ t' \to_\beta t'' \wedge isvalue\ t' = false \implies$ `False` (*IHn*). We again make use of Progress Theorem 1 specialized by the hypothesis (*Ha*), and obtain $isvalue\ t = true \vee \exists e, t \to_\beta e$. We now destruct this fact, and are supposed to prove the goal, which is `False` here, twice assuming $isvalue\ t = true$ and $\exists e, t \to_\beta e$ independently:

    *   $isvalue\ t = true$. It is obvious in this case that the term $t$ does not reduce even a single step further. Namely, there is no $t'$, such that $evaln\ t\ (S\ m) = t'$, which contradicts the hypothesis (*Hb*), and closes the goal.
    *   $\exists e, t \to_\beta e$. We know that there is some term $e$ into which the term $t$ reduces in a single beta-step. We also know by the hypothesis (*Hb*) that $t$ reduces into some term $t'$ in $S\ m$ (or $m + 1$) steps. Putting these together, we infer that the term $e$ reduces into the $t'$ in $m$ steps, namely $evaln\ e\ m = t'$. Moreover, we specialize the Preservation Theorem 2 with the hypothesis (*Ha*) and the fact that $t \to_\beta e$ to retain $\vdash e: \tau$ (*Hte*). This time, making use of (*Hte*) and the fact that $evaln\ e\ m = t'$, we put the induction hypothesis (*IHn*) into the following shape: $\nexists t'',\ t' \to_\beta t'' \wedge isvalue\ t' = false \implies$ `False`. Into this, we plug the conjunction of the hypotheses (*Hne*) and (*Hnv*), and obtain a proof of `False`, which literally implies everything.

□

In a Coq implementation, the proof above could be summarized as follows:

```
Theorem soundness: ∀ n t t' T,
  typecheck nil t = Some T ∧ evaln t n  = Some t' → ¬ stuck t'.
Proof. unfold stuck, not intro n.
       induction n as [| m IHn]; intros t' T (Ha, Hb) (Hne, Hnv).
       - ... (*1*)
         specialize (progress t T Ha); intros Hp. (*Theorem 3.1*)
         destruct Hp as [ Hp | Hp].
       + ...
         contradict Hnv; easy.
       + ...
         contradict Hne; easy.
       - ... (*2*)
         specialize (progress t T Ha); intros Hp. (*Theorem 3.1*)
         destruct Hp as [ Hp | Hp].
       + ... (*2-bullet_1*)
         contradict Hb; easy.
       + ... (*2-bullet_2*)
         specialize (preservation t e T (conj Ha He)); intros. (*Theorem 3.2*)
         specialize (IHn e t' T (conj H Hb)).
         apply IHn. split; easy.
Qed.
```

## 4. Discussion

Observe that the detailed interpreter was developed in an extensible fashion. Namely, it is possible to extend the language of types, such that the object language becomes polymorphically typed. One can similarly extend the language of terms with other programming constructs, such as pairs, lists, match constructs, etc., to handle a richer object language safely interpreted. The formalization already contains an extension of $\lambda^{\rightarrow}$ interpreted as an application. For a use case that deals with a more involved and richer language, we kindly ask the reader to refer to a definitional interpreter (Section 1.1) for a polymorphically typed functional language written in Haskell. The same approach could easily be followed in the current Coq formalization (with additional cases brought over to prove) and reserved to be targeted as future work.

Obviously, for an object language that supports dependent types, the presented approach would not work well. That is because in a dependently typed setting, type-checking requires an evaluation; thus, it is better to have a single language for terms and types.

Remark also that interpreting object languages with computational side effects is beyond the scope of the implementation presented here. It is possible to extend the interpreter's formalization in Coq with certain monads, as in [29], and handle related impurities. Moreover, monads could benefit from generating fresh variables to avoid 'variable capture' that potentially comes up in substituting Lambda terms.

## 5. Conclusions

We developed (in the proof assistant Coq) a definitional interpreter and a type-checker for a purely functional language based on (some extension of) the simply typed Lambda calculus. We formally prove that the type-checker is sound with respect to the evaluation strategy. Namely, every term that type-checks also evaluates (or reduces) unless it is a value. We formalized the soundness proof by employing progress and presentation properties, and discuss the related lemmata in technical detail in the pen-and-paper style containing pointers to the corresponding Coq code. Moreover, there are a few items that could extend the development and be set as future goals. We list them below.

- The formalization does not handle de Bruijn indices [26] or other methods [27,28] that help avoid the variable capture, concerning terms involving free variables. The technical machinery related to these methods will be implemented.
- Extending the interpreter to handle polymorphically typed Lambda calculus embarking on the same approach presented here (a definitional interpreted for a polymorphically typed functional language coded in Haskell) is another goal to achieve.
- The interpreter could be expanded and scaled to handle some other programming blocks, such as *pair*s, *match-end* constructs, *let* bindings, *pairs*, *list*s, and *record*s.

The accompanying Coq sources were tested to compile with `coqc` version `coq-8.15.2` within approximately 43 s on an Intel Core i7-7600U machine running Ubuntu 22.04 LTS over 16 GB of external memory.

**Funding:** This research received no external funding.

**Data Availability Statement:** The data presented in this study are openly available at https://github.com/ekiciburak/extSTLC (accessed on 4 September 2022).

**Conflicts of Interest:** The author declares no conflict of interest.

## Appendix A. Organization of the Coq Sources

To declare terms and types, we employ Coq inductive predicates. As mentioned earlier, type-checking and beta-reduction were implemented, embarking on the definitional approach. The return types of both definitions are wrapped by Coq's `option` type to handle ill-typed terms and those do not reduce any further. Some relevant Coq files from the library are itemized below, such that the subsequent file depends on the previously stated ones. For instance, `Soundness.v` depends on all files.

- Auxiliaries.v: includes some simple proofs of statements about contexts that are indeed defined as lists of pairs.
- Terms.v: involves declarations of terms and types along with decidable equality among terms and types, and reflection proofs of such equalities into Coq's Prop.
- Typecheck.v: the file in which the function `typecheck` is implemented. It also contains proofs of some properties given in Lemmata 1–5. To exemplify how the `typecheck` function works, we implemented the `factorial` function as follows:

```
Definition factorial ≜
  Lambda "f" (Arrow Int Int)
    (Lambda "x" Int
      (ITE (Gt (Ident "x") (NVal 1))
           (Mult (Ident "x") (App (Ident "f") (Minus (Ident "x") (NVal 1))))
           (NVal 1))).
```

One could run the `typecheck` function on the term Fix `factorial` under the empty context (`nil`) with the `Compute` vernacular, and monitor the output that is commented out in the below snippet:

```
Compute typecheck nil (Fix factorial).  (* = Some (Arrow Int Int) *)
```

Of course, the below computation returns `None` as the term is ill-typed under the empty context:

```
Compute (typecheck nil (App (NVal 5) (ITE (NVal 3) (NVal 5) (NVal 10)))).
  (* = None *)
```

- Eval.v: includes the single-step and multi-step beta-reduction functions, respectively, named `beta` and `evaln`. Observe that in exactly 40 steps, the `factorial` function computes the value of `factorial 7` to be 5040:

```
Compute (evaln (App (Fix factorial) (NVal 7)) 40).  (* = Some (NVal 5040) *)
```

Unlikely, the below computation,

```
Compute (beta (App (NVal 5) (ITE (NVal 3) (NVal 5) (NVal 10)))). (* = None *)
```

returns `None` as the input term is stuck.
- Progress.v: contains the proof of Progress Theorem 1.
- Preservation.v: includes the proof of Preservation Theorem 2.
- Soundness.v: contains what it means for a term being stuck (Definition 6) alongside the proof of Soundness Theorem 3.

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
