# Peer review of "A Sound Definitional Interpreter for a Simply Typed Functional Language"

_axioms, doi:10.3390/axioms12010043_

Round 1

Reviewer 1 Report

 Let usconsider that the authors failed to point out any geometrical interesting content and that the paper is just an exercise as for others, therefore I recommend major revision. Because they need to add some extra literature and results.

Author Response

Dear Sir/Madam, first of all many thanks for your comprehensive and constructive comments. We tried our best to extend the paper with some number of points as liseted below:
1. We have extended the introduction section in such a way that it is intended to be more informative about the research presented in the paper.
2. We tried to highlight the contributions simply by itemizing them in the corresponsing sub-section.
3. We have included a discussion section where "potential extensions of the current formalization in a comparative manner with those existing out there" is expressed. A use case has also been referred.
4. We have extended the conclusion with potential future directions.
5. We have also added some number of new references.
We walked trough the paper to fix a few English typos here and there.

Reviewer 2 Report

Paper deals with important tasks of singularities of the Interpreter for a Simply Typed Functional Language. 

It has a logical structure. However, I have a number of suggestions:   

1. The authors could consider adding to the "Introduction" section the motivation for this research, i.e., the research need and importance.
2. I would suggest removing all the sources from the paper, and providing access to them in the GitHub repo or something like that.

3. Authors should clearly point-by-point describe the main contributions of this paper. It should somehow resonate with the title of the work.

4. Proposed solution isn't unclear and isn't possible to realize its use cases. I would suggest providing a better visual description (performance comperison) and architecture approach presentation, with usecases presentation.

5. The paper hasn’t any Discussions.
6. The conclusion section should be extended with prospects for future research.

7. A lot of references are outdated and unlinked. Please fix it by using 3-5 years old papers in high-impact journals

Author Response

Dear Sir/Madam, first of all many thanks for your comprehensive and constructive comments. We tried our best to address every single point you have indicated as listed below:

1. We have extended the introduction section in such a way that it is intended to be more informative about the research presented in the paper.

2. Removing the sources from paper may break the structure as we have written proofs on the paper and also in Coq together with a bridge across them. This way, readers could commute and observe the one to one correspondence between a Coq proof and its paper reflection. The entire Coq source could be accessible at a github repo to which a reference has been
given in the paper.

3. We tried to highlight the contributions simply by itemizing them in the corresponding sub-section.

4.-5. We have included a discussion section where "potential extensions of the current  formalization in a comparative manner with those existing out there" is expressed. A use case has also been referred.

6. We have extended the conclusion with potential future directions.

7. We have also added some number of new references.

We walked trough the paper to fix a few English typos here and there.

Round 2

Reviewer 1 Report

I accept the revised version

Reviewer 2 Report

old version of the article uploaded again :(